



# Optical Properties and Shape-Dependent Complex Refractive Index Retrievals of Freshly Emitted Saharan Dust

Jesús Yus-Díez[1,2,*], Jeronimo Escribano[3,*], Marco Pandolfi[1], Andrés Alastuey[1], Cristina González-Flórez[3,a], Adolfo Gonzalez-Romero[3], María Gonçalves Ageitos[3,8], Matic Ivančič[4], Martina Klose[5], Konrad Kandler[6], Vincenzo Obiso[3], Agnesh Panta[6,b], Cristina Reche[1], Martin Rigler[4], Xavier Querol[1], and Carlos Perez García-Pando[3,7]

[1]Institute of Environmental Assessment and Water Research (IDAEA-CSIC), Barcelona, 08034, Spain
[2]Center for Atmospheric Research, University of Nova Gorica, Vipavska 11c, Ajdovščina, 5270, Slovenia
[3]Barcelona Supercomputing Center (BSC), Barcelona, Spain
[4]Aerosol d.o.o., Ljubljana, Slovenia
[5]Karlsruhe Institute of Technology (KIT), Institute of Meteorology and Climate Research - Troposphere Research (IMK-TRO)
[6]Technical University Darmstadt (TUDa), Darmstadt, Germany
[7]ICREA, Catalan Institution for Research and Advanced Studies, Barcelona, 08010, Spain
[8]Universitat Politècnica de Catalunya, Barcelona, Spain
[*]These authors contributed equally to this work.
[a]now at: Danish Meteorological Institute, Copenhagen, Denmark
[b]now at: Palas GmbH, 76187 Karlsruhe, Germany

**Correspondence:** Jesús Yus Díez (jesus.yus@ung.si) and Jerónimo Escribano (jeronimo.escribano@bsc.es)

**Abstract.**

Mineral dust is a major contributor to atmospheric aerosols and plays a complex role in Earth's radiation budget. However, large uncertainties remain in quantifying its direct radiative effects (DRE), primarily due to poorly constrained absorption properties. Most estimates rely on remote sensing or laboratory studies, with few in situ measurements of freshly emitted

5    dust. This study presents field data on dust optical properties collected during active dust emission in the Moroccan Sahara (September 2019). During high emission events, optical properties aligned with previous Saharan dust studies. Single scattering albedo (SSA) for PM2.5 (PM10) was 0.95 (0.94) at 370 nm and 0.97 (0.96) at 660 nm. Coarse particles contributed to negative scattering and SSA Ångström exponents (SSAAE), and absorption Ångström exponents (AAE) reached up to 2.5 (2.0), indicating strong wavelength-dependent absorption from iron oxides. The asymmetry parameter (g) was 0.7 (0.65) at 520

10   nm, while backscatter fraction (BF) was 0.11 (0.13), showing coarse dust's impact on scattering.-Mass absorption efficiency (MAE) decreased from  0.30 to 0.15 $m^2 \, g^{-1}$ at 370 nm with increasing particle size. Mass scattering efficiency (MSE) shifted toward longer wavelengths for larger particles. A key result is our consistent retrieval of the imaginary refractive index (k) across wavelengths, accounting for dust's irregular shape. Retrieved k increased linearly with particle asphericity, rising from 0.0011 for spheres to 0.0016 for triaxial ellipsoids at 520 nm—a  60% enhancement. These findings highlight the need for

15   realistic and consistent particle shapes and k in satellite retrievals and climate models.



# 1 Introduction

Mineral dust emitted from arid regions is one of the most abundant aerosol types by mass (Myhre et al., 2013) and influences the Earth's climate through multiple mechanisms. It exerts a direct radiative effect (DRE) by absorbing and scattering solar and terrestrial radiation (Kok et al., 2017; Adebiyi and Kok, 2020), a semi-direct effect by modifying the atmospheric temperature profile and influencing cloud properties (e.g., Ackerman et al., 2000; Perlwitz and Miller, 2010), and an indirect effect by altering cloud microphysics and precipitation patterns (Karydis et al., 2017; McGraw et al., 2020). Additionally, mineral dust has significant impacts on the cryosphere (Skiles et al., 2018) and atmospheric chemistry (Mahowald, 2011), contributing to climate and environmental changes (Kok et al., 2023).

Despite its importance, the DRE of mineral dust remains highly uncertain, primarily due to poorly constrained dust absorption properties (Samset et al., 2018). Current estimates place the short-wave DRE at a central value of $-0.40 \pm 0.30 \ \mathrm{Wm}^{-2}$ (90% confidence) based on model assessments that integrate dust optical depth, optical properties and particle size distributions (PSDs) consistent with limited experimental constraints (Kok et al., 2023). Furthermore, Kok et al. (2023) found a 55 $\pm$ 30% increase in global dust aerosol mass concentrations since pre-industrial times and additional uncertainties on other dust interactions, with an associated global negative mean effective radiative forcing of approximately $-0.07 \ \mathrm{Wm}^{-2}$, partially offsetting greenhouse warming. However, the uncertainty in this estimate remains substantial ($\pm 0.18 \ \mathrm{Wm}^{-2}$), leaving open the possibility that dust aerosols may have contributed to net warming since pre-industrial times (Kok et al., 2023).

Key dust characteristics influencing its optical properties—and consequently its direct DRE include variations in mineralogical composition, PSD, particle shape and morphology, and mixing state (e.g. Kok et al., 2017; Di Biagio et al., 2020; Adebiyi and Kok, 2020; Li et al., 2021). Uncertainties in these properties, along with uncertainties in the total atmospheric dust mass and its vertical distribution, significantly impact the accuracy of DRE quantification.

Dust optical properties, particularly the complex refractive index (CRI $= n + i\,k$), play a critical role in determining the dust radiative impact. The real part of the CRI ($n$) predominantly determines scattering, while the imaginary part ($k$) controls absorption. However, the imaginary component also influences scattering because a higher absorption will, for a given extinction, diminish the scattering coefficient. The intrinsic CRI of dust depends on its mineralogical composition and describes how a homogeneous material interacts with light at a microscopic scale. For instance, the presence of iron oxides such as goethite and hematite, plays a dominant role in dust absorption (e.g., Di Biagio et al., 2019). Several studies have also found that the relative fraction of total iron in dust correlates with its absorption properties, influencing radiative forcing estimates (Sokolik and Toon, 1999; Lafon et al., 2006; Balkanski et al., 2007; Caponi et al., 2017). The imaginary part of the CRI for iron oxides exhibits increased absorption at short-wavelength UV compared to red and near-infrared wavelengths (e.g. Sokolik and Toon, 1999; Balkanski et al., 2007). However, particle size and shape influence how a given CRI translates into observable optical properties. It is important to note that in real-world measurements, the effective CRI retrieved from bulk optical observations depends not only on composition but also on the particle size and shape distribution of the dust particles. This is because dust is composed of irregularly shaped particles spanning a wide range of sizes, and retrievals from optical measurements inherently integrate these variations (Gasteiger and Wiegner, 2018, e.g.,).



The PSD of mineral dust exerts a strong influence on its optical properties, as the size parameter (ratio of particle circumference to wavelength) dictates the dominant scattering regime (Mishchenko et al., 2002). Smaller particles often display more uniformly distributed scattering angles, whereas larger particles tend to scatter light strongly in the forward direction, thereby increasing total extinction but decreasing backscattering (e.g., Sokolik and Toon, 1999). Coarse-mode dust can dominate dust mass and exert substantial effects on direct radiative forcing, yet is often under-represented in models (Adebiyi and Kok, 2020). Since optical measurements are sensitive to both the particle size distribution and its interaction with light, retrievals of the CRI inherently reflect the size-dependent scattering and absorption behaviour of dust.

Particle shape further modulates optical properties. Several studies have reported that total extinction from non-spherical dust particles deviates from spherical models by roughly 10–20% (e.g., Nousiainen, 2009; Bi et al., 2018), especially when averaged across a typical size distribution. However, Kok et al. (2017) showed that shape effects on extinction could even be significantly larger. Despite these potential differences in total extinction, the single scattering albedo (SSA) often remains relatively unchanged or only modestly affected. Conversely, shape tends to have a more pronounced impact on the scattering phase function and asymmetry factor (g), influencing how light is directionally scattered. Retrievals of the effective CRI are impacted by shape assumptions, as standard inversion algorithms often rely on Mie theory, which assumes sphericity. The use of spherical models can introduce systematic biases, particularly in the retrieved imaginary part of the CRI, which governs absorption.

Thus, while the intrinsic CRI is composition-dependent, the effective CRI derived from measurements is shaped by the interplay of mineralogy, PSD, and non-sphericity. Additional complexities are originated from internal mixing and physicochemical ageing of dust with other aerosol species during transport. This distinction is crucial for interpreting observational data, improving remote-sensing retrievals, and reducing uncertainties in climate models.

Climate models require accurate representations of dust optical properties, including the complex refractive index, single-scattering albedo, and asymmetry factor. Some studies have examined how various mineralogical compositions and PSDs affect these parameters, often through laboratory experiments with resuspended soil samples (e.g., Di Biagio et al., 2017; Di Biagio et al., 2019), ground-based or satellite remote-sensing, or in-situ observations in known dust outflow regions. Indeed, direct in-situ measurements of freshly emitted dust within source regions remain relatively scarce. Most existing observations in deserts focus on campaigns that capture dust outflow rather than the exact point of emission (Kandler et al., 2009; Müller et al., 2009; Schladitz et al., 2009), leaving gaps in understanding how local soil characteristics and emission processes govern the initial PSD and optical properties of dust. Measurements from large networks such as AERONET also provide valuable remote retrievals of total-column aerosol properties, though the presence of other particles can complicate dust-specific analyses (e.g., Obiso et al., 2024).

To address the gap on the local emission processes and its relationships, a field campaign was conducted in the Saharan desert (Morocco) in September 2019 under the framework of the FRAGMENT project. One major aim was to characterize the size-resolved mineralogical composition of freshly emitted dust and the emission mechanisms that inject these particles into the lower troposphere (González-Romero et al., 2023, 2024a, b; Panta et al., 2023; González-Flórez et al., 2023). Another





goal was to measure and interpret the multi-wavelength optical properties—specifically, scattering and absorption, which is the focus in the present study.

Here, we present high-temporal-resolution measurements of extensive properties (i.e., scattering and absorption coefficients) and derived intensive properties, including SSA, asymmetry parameter, backscatter fraction, and scattering, absorption and extinction efficiencies at several wavelengths of freshly emitted dust at a key Saharan source region. We further link these observations to the size of the emitted particles and use aspect-ratio information derived from single particle analyses (Panta et al., 2023) together with optical particle counter measurements (González-Flórez et al., 2023) to retrieve the imaginary part of the CRI ($k$). The influence of different shape assumptions on the retrieval of $k$ is also explored. These high-resolution, source-region measurements provide crucial insights for improving remote-sensing algorithms and reducing uncertainties in climate models (e.g., Castellanos et al., 2024).

In the following, Sect. 2 presents the measurement area, the instruments used, and the key optical properties examined in this study, highlighting their relevance for mineral dust. It also provides a detailed explanation of the methodology employed to estimate $k$. Sect. 3 then presents the findings in three parts: (i) the temporal evolution of the measured optical properties, focusing on three representative scenarios that are particularly important for understanding dust emission into the atmosphere; (ii) the relationships between these optical properties and the size of the measured aerosol particles; and (iii) the retrieved values of $k$ based on the above measurements .

## 2 Methodology

### 2.1 Measurement site

In-situ dust measurements were conducted in a mineral dust emission area in southeastern Morocco, at the transition between the southern foothills of the Atlas Mountains and the Saharan Desert. The study site is located in L'Bour (29° 49' 30" N, 5° 52' 25" W, 500 m a.s.l.), a small terminal depression near the Drâa dry riverbed, west of M'Hammid El Ghizlane. It is also situated east of Lake Iriki, a seasonal dry lake, and adjacent to two major dune fields, Erg Cheggaga and Erg Smar (see Fig. 1).

The surface at L'Bour is predominantly hard-packed, featuring crust formations resulting from wind and dust erosion, interspersed with shallow sand dunes (height <1 m). These characteristics make it a representative site for studying freshly emitted dust from natural sources. Further details on the sampling site and its environmental conditions can be found in González-Flórez et al. (2023); González-Romero et al. (2023); Panta et al. (2023).

### 2.2 Measurements of optical properties

Multi-wavelength aerosol absorption coefficients ($b_{abs}^{\lambda}$) were derived using a multi-wavelength dual-spot aethalometer (AE33, Aerosol d.o.o.]). The AE33 measures light attenuation at seven wavelengths spanning from the short-UV to the near-infrared range (370, 470, 520, 590, 660, 880, and 950 nm). Absorption coefficients are then computed using equation (17) in Drinovec et al. (2015), with an online correction for filter loading effects applied by the AE33 software.



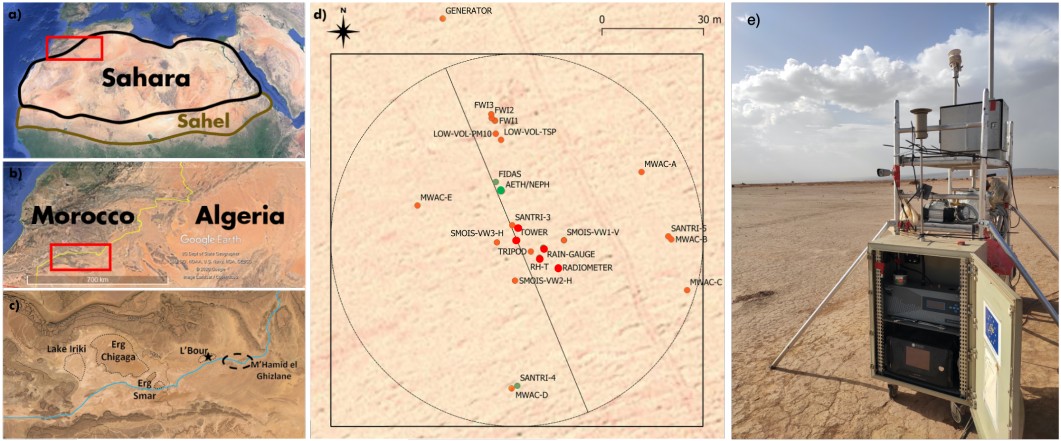

**Figure 1.** Location of the area of study over Northern Africa (a,b and c,), the instrument deployment over the L'Bour site, where the diagonal black line is perpendicular to the approximate predominant wind direction estimated based on prior data analysis (d), the green points correspond to the optical instruments (AE33 and nephelometer) and the Fidas, which are shown in (e), where the optical instruments are placed in cage in front of the scaffolding and the Fidas is located in the background behind the scaffolding. The integrating nephelometer is placed in the upper level and the Aethalometer AE33 in the lower one. Satellite imagery in panels a, b, c and d were taken from base layer from world imagery of © Google Earth Pro v:7.3.6.9345. The photo in panel e) was taken by one of the authors.

To obtain accurate absorption coefficients from AE33 measurements, an additional correction for multiple light scattering within the filter tape was applied. The multiple-scattering parameter, C, has been shown to be higher for aerosols with high single scattering albedo, such as mineral dust, compared to typical values found in urban and regional background sites (e.g. Di Biagio et al., 2017; Yus-Díez et al., 2021). Since measurements were performed in an environment dominated by dust particles, we used a C value obtained by optimizing SSA and C as recommended in Yus-Díez et al. (2021), using the C and multiple

scattering fitted values provided therein in a mountain-top station during Saharan dust outbreaks. The obtained C value from the fit is 4.82, which is on the upper limit of those obtained for dust by Di Biagio et al. (2017).

Scattering coefficients ($b_{scat}$) were measured using an LED-based multi-angle integrating nephelometer (Aurora 4000, ECOTECH Pty Ltd, Knoxfield, Australia), which operates at three wavelengths (450, 525 and 635 nm) and collects data at eight angular positions (0°, 10°, 25°, 40°, 55°, 70°, 90°, and 170°). Forward (0°) and backward (170°) scattering measure-

ments were then corrected for non-ideal illumination of the light source and truncation errors using the correction scheme from Müller et al. (2011) for coarse particles.

During the initial phase of the L'Bour field campaign (2019-09-04 00:00 UTC to 2019-09-26 12:00 UTC), in-situ surface optical measurements were performed using a PM2.5 inlet. After this date and until the end of the campaign (2019-10-01 08:00:00 UTC), PM10 sampling was used instead. To reduce the instrumental noise while accounting for all significant turbulent struc-

tures carrying momentum flux (Dupont et al., 2019), all the campaign measurements, including the optical measurements, were averaged into 15 min intervals for analysis.



## 2.3 Particle size distribution measurements

Number PSDs were measured using a Fine Dust Aerosol Spectrometer (Fidas 200S, Palas GmbH). This instrument sampled aerosol particles across 63 bins spanning optical diameters of equal logarithmic width from 0.20 to 19.1 μm with a 120 s time resolution and an inlet flow rate of $4.8 \, \mathrm{l\,min^{-1}}$. Data from the first three bins were not used as they were found to be unrealistic (cf. González-Flórez et al., 2023). Similar to the optical measurements, data was averaged into 15 min intervals.

The Fidas instrument was calibrated in the field using polystyrene latex spheres (PSLs) with a CRI of $1.59 + 0i$. To derive a PSD that better represents dust particles in terms of their geometric diameter, we applied a diameter conversion procedure as described in Sect.2.6.1 and based on the method outlined by Huang et al. (2020). This conversion requires assumptions about the particles' CRI and shape. In Sect.3.2, we analyze intensive optical properties. For those depending on particle mass, we use the converted PSD assuming spherical particles and an initial averaged CRI guided by Di Biagio et al. (2019). These initial estimates correspond to CRIs with real parts of 1.47, 1.50, and 1.52, and imaginary parts ranging from 0.0001 to 0.005. The initial assumptions are subsequently refined through an inversion procedure described in Sect.2.6, where the dust imaginary CRI is consistently retrieved and further analyzed in Sect.3.3.

From the resulting number PSD, we calculated the mass concentration ($\mathrm{M_{PM}}$) and the effective radius ($\mathrm{R_{eff}}$). $\mathrm{M_{PM}}$ is given by the third moment of the number PSD,

$$\mathrm{M_{PM}} = \rho \cdot \frac{\pi}{6} \cdot \int_{\mathrm{D_0}}^{\mathrm{D_x}} \mathrm{D^3 N(D) dD};$$

$$(1)$$

where D is the particle geometric diameter, and $\mathrm{N(D)}$ is the dust number PSD for diameter D derived from the Fidas, and $\rho$ is the particle density ($2500 \, \mathrm{kgm^{-3}}$ as in González-Flórez et al., 2023). $\mathrm{R_{eff}}$ is then defined as the ratio of the third to the second moment of the number PSD,

$$\mathrm{R_{eff}} = \frac{1}{2} \cdot \frac{\int_{0.2}^{\mathrm{D_x}} \mathrm{D^3 N(D) dD}}{\int_{\mathrm{D_0}}^{\mathrm{D_x}} \mathrm{D^2 N(D) dD}};$$

$$(2)$$

Here, D spans the relevant diameter range between the first used bin $D_0$ and $D_x$. $D_x$ is the geometric diameter corresponding to the aerodynamic diameter of the inlet cut-off, which will vary depending on the inlet cut-off period –PM2.5 or PM10. The conversion from optical to geometric diameters from the Fidas is explained in Sect. 2.6.1

## 2.4 Meteorological measurements

We used wind measurements that were conducted at 2 m, approximately matching the inlet heights of the optical and PSD instruments, with a 2-D sonic anemometer (model WINDSONIC4-L, Campbell Scientifics, USA) mounted on a 10 m meteorological tower. As detailed in González-Flórez et al. (2023), this tower also hosted multiple 2-D and 3-D sonic anemometers,





with the data from these instruments undergoing extensive processing to derive parameters such as wind friction velocity ($u_*$).
Briefly, $u_*$ was obtained via an iterative law-of-the-wall approach, which assumes a (pseudo)logarithmic wind profile in the atmospheric surface layer (cf. González-Flórez et al., 2023, and references therein). Although the raw wind data were recorded at 0.5 Hz, they were averaged to 15 min intervals during post-processing to facilitate a stable, consistent comparison with the other collocated instruments.

## 2.5 Intensive aerosol optical properties

By combining the extensive optical properties (absorption from the AE33 and scattering from the polar nephelometer) with the PSDs from the Fidas, we derived time series of intensive optical properties. These intensive optical properties depend on the physico-chemical properties of the dust particles, including mineralogy, PSD, and shape. Below is a brief description of each parameter:

a. The Scattering Ångström Exponent (SAE) mostly reflects particle size, with values less than 1 typically indicating
coarse particles (Seinfeld and Pandis, 1998; Schuster et al., 2006), as often observed during fresh dust emissions in our campaign. The SAE was calculated by fitting the scattering coefficients (at three nephelometer wavelengths) in a log-log space

$$\ln(\mathrm{b_{scat}}(\lambda)) = -\mathrm{SAE} \cdot \ln(\lambda) + \beta. \tag{3}$$

b. The Absorption Ångström Exponent (AAE) is strongly influenced by the composition of the sampled particles. Under
175 anthropogenic-dominated conditions the AAE typically ranges from 0.9 to 1.1 (Kirchstetter et al., 2004; Petzold et al., 2013), whereas dust-rich environments yield values between 2 and 6.5 (Schuster et al., 2016; Caponi et al., 2017). This higher AAE stems from dust enhanced absorption in the short-UV region relative to near-infrared wavelengths (Kirchstetter et al., 2004; Chen and Bond, 2010). This spectral behavior of the absorption efficiency of the particles is attributed to the optical properties of the iron oxides contained in mineral dust (e.g. Schuster et al., 2006; Müller et al.,
2009; Schladitz et al., 2009). Here, we derived the AAE using a similar log-log fit, but applied to multi-wavelength absorption from the AE33.

$$\ln(\mathrm{b_{abs}}(\lambda)) = -\mathrm{AAE} \cdot \ln(\lambda) + \beta. \tag{4}$$

c. The Single Scattering Albedo (SSA) calculated as the ratio of the scattering ($\mathrm{b_{scat}}$) to the extinction ($\mathrm{b_{ext}} = \mathrm{b_{scat}} + \mathrm{b_{abs}}$), $SSA = b_{scat}/b_{ext}$ coefficients at all the AE33 wavelengths, is a crucial parameter in climate modeling because it indi-
185 cates whether aerosols predominantly scatter (cooling effect) or absorb (warming effect) radiation. Typical values for Saharan dust range from 0.92 at 370 nm to 0.98 at 880 nm (Di Biagio et al., 2019), and from 0.90 to 0.96 at the green wavelengths depending on the estimated dust iron content (Claquin et al., 1999; Schladitz et al., 2009). We computed the SSA at the seven AE33 wavelengths by extrapolating the nephelometer scattering data using the derived SAE.



d. The Single Scattering Albedo Ångström Exponent (SSAAE) helps identify coarse particles (e.g. dust) with values below zero often tied to dust-dominated conditions (Collaud Coen et al., 2004; Ealo et al., 2016). We obtained the SSAAE by fitting the wavelength dependence of the derived SSA values (at the AE33 wavelengths) in a log-log space, analogous to the SAE/AAE approach.

$$\ln(\mathrm{SSA}(\lambda)) = -\mathrm{SSAAE} \cdot \ln(\lambda) + \beta. \tag{5}$$

e. The asymmetry parameter (g) is the first moment of the Legendre decomposition of the scalar scattering phase function. It ranges between $-1$ and $1$, where a value of $-1$ indicates pure backward scattering and a value of $1$ refers to pure forward scattering. For airborne dust particles, values between 0.6 and 0.85 are commonly used in numerical models (Sokolik and Toon, 1999; Horvath et al., 2018; Ryder et al., 2018). Here we have obtained $g$ from the phase function of the nephelometer multi-angle measurements, corrected by angular truncation as indicated in Sect. 2.2.

$$\mathrm{g} = \frac{1}{2} \int_{-1}^{1} \mathrm{P}(\cos\theta)\,\cos\theta\,\mathrm{d}\cos\theta, \tag{6}$$

where $\mathrm{P}(\cos\theta)$ is the phase function (Liou, 2002; Horvath et al., 2018).

f. The Backscatter Fraction (BF) is computed as the ratio between the hemispheric backward scattering and the total scattering from the corrected nephelometer measurements.

g. The Mass Absorption, Scattering, and Extinction efficiencies (MAE, MSE, and MEE) represent the absorption, scattering, and extinction efficiencies, respectively, of the measured particles per unit of mass. The MAE, MSE and MEE were obtained as the ratio between the absorption, scattering, and extinction coefficients, respectively, and the mass concentration derived from the number PSD ($\mathrm{M_{PM}}$) which, given the location of the measurements site, can be attributed exclusively to mineral dust during the strong mineral dust emission events. Values of dust MEE around $0.32\,\mathrm{m^2\,g^{-1}}$ have been reported in previous studies (Ryder et al., 2018).

## 2.6 Complex refractive index retrieval

The imaginary part of the complex refractive index ($k$) was estimated using a combination of optical property measurements and absorption and scattering simulations based on the measured PSDs. In particular, we determined the value of $k$ that minimizes the discrepancy between simulated and measured SSA. This optimization was performed under different assumptions regarding particle shape.

Our optimization methodology builds on the approach of Di Biagio et al. (2019) but introduces key modifications, particularly in its minimization structure. The process consists of three main steps: (1) estimating the number PSD in terms of dust



geometric diameters, which depends on both particle shape and $k$; (2) simulating scattering and absorption coefficients based on the measured number PSD; and (3) minimizing a quadratic cost function based on SSA.

A key difference from Di Biagio et al. (2019) is that our optimization of $k$ accounts for the interdependence between the PSD diameter conversion and the simulated optical coefficients. Since the transformation of the PSD from PSL-based optical diameters to dust geometric diameters depends on $k$, and the absorption coefficient is directly influenced by $k$, both must be optimized simultaneously. Therefore, for a given value of $k$, the scattering and absorption coefficients, and the size-resolved phase function are computed. The phase function and the scattering coefficients are used in the PSD conversion; while the integration of the phase function, scattering and absorption coefficients are combined with the converted PSD to simulate the measured absorption and scattering coefficients. This procedure ensures full consistency in the size distribution transformation and the simulated optical properties.

Another key difference from the Di Biagio et al. (2019) method is that we define the cost function directly in terms of SSA, as this parameter is highly sensitive to $k$. Although we tested alternative cost functions incorporating extensive information from the polar nephelometer, our dataset did not support a robust simultaneous retrieval of both $n$ and $k$. This limitation likely stems from the sensitivity of $k$ to particle shape assumptions, inlet cut-off modeling, and PSD uncertainties. To address this, we retrieved $k$ for multiple fixed values of $n$ rather than solving for both parameters simultaneously. As demonstrated in Sect. 3, the retrieved $k$ remains largely insensitive to the choice of $n$ within its plausible range, reinforcing the reliability of our approach. Below we describe the three steps in detail.

### 2.6.1 Estimation of the dust PSD

The number PSD obtained from the Fidas instrument is calibrated using polystyrene latex spheres (PSL, CRI = 1.59 + 0i), requiring conversion to dust geometric diameters. As described in González-Flórez et al. (2023), we transform the default PSL diameters into dust geometric diameters following Huang et al. (2020), which involves calculating the theoretical scattered intensities of the PSLs and those of the aspherical and absorbing dust. The comparison of both sideward scattered intensities allows remapping the PSL into dust geometric diameters after a monotonization procedure for ensuring uniqueness of the solution. The calculation of the scattered intensities depends on the wavelength of the light beam used in the Fidas, the scattering angle range of the OPC's light sensor, and the shape and CRI of the particles. We first inferred the scattered light spectrum of the Fidas instrument (which is not provided by the manufacturer) by analyzing its manufacturer's conversion software, which translates PSL-based PSDs to those of spherical aerosols with 16 different refractive indices. Using the known $90° \pm 5°$ scattering angle and a Lorenz-Mie scattering model, we optimized a Gaussian spectral function that best reproduced these conversions, yielding an estimated light spectrum centered at 389 nm with a 77 nm standard deviation. This optimized spectrum was then applied to transform PSL optical diameters into dust geometric diameters.

The sideward scattered intensity depends on particle shape. Since PSLs are spherical, we obtained their single-scattering properties based on Lorenz–Mie theory. For dust, we used a wide range of particle shapes available in the MOPSMAP database (Gasteiger and Wiegner, 2018) to assess the impact of shape assumptions on the retrieval of $k$. We tested spherical particles and biaxial prolate ellipsoids with aspect ratio of $1.46$, along with a distribution of biaxial prolate ellipsoids based on aspect ratios



measured during our campaign using Scanning Electron Microscopy single particle analysis (Panta et al., 2023). Additionally, we included six irregular dust particle shapes (A to F) defined in Gasteiger et al. (2011) and MOPSMAP (Gasteiger and Wiegner, 2018). Beyond these, we incorporated triaxial ellipsoids from Meng et al. (2010), selecting also $1.46$ as aspect ratio (Panta et al., 2023), and a height-to-width ratio of $0.45$. This choice aligns with the median height-to-width ratio ($0.4$) reported in Huang et al. (2020) and represents the closest available match in the Meng et al. (2010) database for our measured size 255 range and wavelengths. By testing this diverse set of shape assumptions, we ensure a more comprehensive assessment of morphological effects on the derived optical properties.

### 2.6.2 Scattering and absorption simulations

Using the dust PSD as input, we computed scattering and absorption coefficients using the single-particle optical property databases for each shape assumption from Gasteiger and Wiegner (2018) and Meng et al. (2010). To accurately reproduce the 260 nephelometer's measurement characteristics, we simulated its angular sensitivity and truncation effects, applying the correction method from equation (11) of Teri et al. (2022). These computations provided the truncated scattering coefficient and the absorption coefficient. To avoid introducing additional assumptions, we defined a modified SSA, using the truncated scattering coefficient rather than an assumed full-phase-function scattering coefficient. Finally, a quadratic cost function was used to compare the simulated and observed SSA, with $k$ optimized via grid-search minimization as in Di Biagio et al. (2019).

We note that, while the aethalometer and nephelometer measurements were taken using a PM2.5 inlet, the PSD measurements used an inlet with a larger diameter cut-off (González-Flórez et al., 2023). In order to use the PSD measurements to simulate to the optical property measurements during the $k$ retrieval algorithm, we also estimated the aethalometer and nephelometer effective inlet cut-off by applying a sigmoid function following the PM2.5 URG model described in Keywood et al. (1999). In addition, we assumed an aerodynamic to geometric diameter ratio for dust of 1.7 (Huang et al., 2020), which ac-270 counts for dust density and shape factor corrections. The sensitivity of the retrieved $k$ to this assumption was later quantified by repeating the whole procedure for and inlet cut-off diameter 10% larger (equal 1.87) and 10% smaller (equal 1.53) than our best guess value of 1.7.

### 2.6.3 Minimization procedure

The minimization was performed independently for each AE33 wavelength, using 15-minute averaged measurements with 275 collocated PSD and optical data. We repeated the optimization for multiple fixed values of $n$ within the 1.48–1.55 range, in 0.01 increments, following the reported values from Di Biagio et al. (2019). To assess particle shape effects, we conducted calculations for the diverse set of particle morphologies described above.

### 2.6.4 Retrieval of CRI from AERONET

To contextualize and compare our retrieved $k$ values, we derived a dust-specific $k$ climatology from the AErosol RObotic 280 NETwork (AERONET) version 3, level 2.0 Almucantar retrievals (Giles et al., 2019; Sinyuk et al., 2020). Dust-dominated





conditions were identified following the filtering methodology of Obiso et al. (2023), which applied a series of criteria to exclude retrievals contaminated by other absorbing aerosols, such as carbonaceous species.

AERONET hourly retrievals were classified as dust events when the fine-mode volume fraction was below 10%, ensuring that the dataset was dominated by coarse particles. Additionally, we required that the SSA increased from 440 to 675 nm,
a spectral trend characteristic of dust that helps distinguish it from fine anthropogenic aerosols and sea salt (Dubovik et al., 2002). To further exclude the influence of absorbing black and brown carbon, we filtered out cases where the mean $k$ at red and infrared wavelengths (675, 870, and 1020 nm) exceeded 0.0042, following the thresholds proposed by Schuster et al. (2016).

Using this filtered dataset, we computed the climatological dust $k$ at the Ouarzazate AERONET station, the closest site to our field measurements (160 km). Additionally, we extended our analysis to major global dust source regions (see Fig. S9)
to assess the broader variability of dust absorption properties. In the regional analysis, we applied optical depth weighting to compute statistical metrics, including mean and percentiles, ensuring that the retrieved $k$ values reflect both temporal and geographical variability in dust retrievals.

## 3 Results

### 3.1 Overview of the optical properties during the measurement campaign

The campaign was characterized by overall calm conditions in the early morning, followed by stronger winds in the afternoon, leading to frequent local saltation and dust emission. This pattern followed the diurnal cycle of solar heating and the resulting destabilization of the near-surface atmospheric layers. Some days showed advection of local anthropogenic particles from up-river villages. Additionally, two intense cold pool outflows, referred to as "Haboobs" were recorded on the 4th and 6th of September 2019. A more detailed discussion of the atmospheric conditions, turbulence parameters, and dust fluxes during the
campaign can be found in González-Flórez et al. (2023).

Two distinct measurement periods were defined based on the inlet cut-off of the optical measurements: an initial period with a PM2.5 inlet from the beginning of the campaign until September 26, followed by a PM10 inlet period from that point onward. Figure S1 presents the temporal evolution of extensive and intensive optical properties, mass concentration, particle effective radius, wind speed, and friction velocity throughout the whole duration of the campaign.

To analyze the dominant aerosol conditions during the measurement period, we identified three representative scenarios: local dust emissions (Period A, 21st of September 2019), Haboob-driven dust events (Period B, 6th of September 2019), and anthropogenic pollution advection mixed with minor local dust emissions (Period C, 7th–10th of September 2019). These periods are highlighted in Figure 2.

Table 1 provides a summary of the time-averaged optical properties for the PM2.5 and PM10 periods, distinguishing between
310 overall measurements, dust events, and the three defined periods. Dust events during the PM2.5 period were identified based on an AAE above 1.5, an SSAAE below 0, and a mass concentration exceeding 10 μg·m$^{-3}$ (Collaud Coen et al., 2010; Ealo et al., 2016). Extensive optical properties are presented with their standard errors, while intensive properties are averaged using two approaches: (i) weighting by extensive properties (absorption, scattering, and extinction for their corresponding variables)



and (ii) direct time-averaging with standard error representation. The weighted averages provide insight into aerosol properties under conditions of elevated dust concentrations, while the direct averages reflect the overall campaign conditions.

Period A, dominated by local dust emissions, closely resembles the overall dust event averages, confirming that such emissions were common throughout the campaign. Period B (Haboob event) exhibits similar intensive optical properties to Period A, though with substantially higher values in extensive optical properties due to increased dust mass concentrations. In contrast, Period C, influenced by anthropogenic pollution, shows lower mass concentrations and therefore reduced absorption and scattering coefficients, yet much higher mass absorption efficiencies, particularly at longer wavelengths. This indicates an enhanced influence of fine carbonaceous particles on the site's optical properties during periods of weak easterly winds and the absence of local dust emissions, as analyzed in González-Flórez et al. (2023).

Figure 2 and Table 1 show that during dust events, SSA was around 0.955 at 370 nm, 0.985 at 550 nm and 0.993 at 880 nm for PM2.5, while the AAE remained near 2 and the SAE remained below 0. Brief decreases in effective radius, accompanied by dips in SSA and increases in SAE and $k$, again suggest intermittent advection of anthropogenic particles to the site under low wind and dust contributions, driven by local meteorology. For PM10, SSA is reduced to around 0.945 at 370 nm, 0.974 at 550 nm and 0.984 at 880 nm (Table 1).

A comparison with other dust SSA measurements in the literature (Table S-1 in the Supplementary Material of Adebiyi et al. (2023)) shows that our PM10 results lie well within the general range of Saharan dust SSA values reported by other in-situ and remote-sensing studies (0.957 at 537 nm in Tinzou from Schladitz et al. (2009) and 0.971 at 550 nm in Western Sahara from Johnson and Osborne (2011)). As expected, the measured PM2.5-based values—focusing on finer particles—tend toward slightly higher SSA at near-infrared wavelengths.

Many of the references in Adebiyi et al. (2023) measure a broader particle size spectrum or long-range transported dust, potentially lowering SSA due to the inclusion of coarser and more absorbing dust particles, whose effect is analyzed in Sect. 3.2. Furthermore, Adebiyi et al. (2023) Table S1 reflects dust sampled across various regions where mineralogy and aerosol mixing states often differ from fresh emissions in southeastern Morocco, with some samples collected far from the source area (in Cabo Verde and the Canary Islands over the Atlantic or in the West Mediterranean). While our measurements capture the optical properties both at PM2.5 and PM10 fractions, they focus on a near-source environment and may not mirror the particle ageing and compositional evolution encountered during trans-Atlantic transport or in other source areas. Consequently, although our SSA measurements overlap with previous studies, differences in size range, sampling region, and dust ageing processes must be considered when comparing results across multiple datasets.

The mass efficiencies, MAE, MSE and MEE, as expected, show a high variation with inlet cut-off—1 order of magnitude—since most of the mass concentration of dust particles is found in the coarse fraction. The MEE found herein for dust events during the PM2.5 period ($0.145 \, \text{m}^2\text{g}^{-1}$ at 375 nm) is higher than the one found in Caponi et al. (2017) for Saharan dust ($0.107 \, \text{m}^2\text{g}^{-1}$ at 375 nm), but lower than dust from the Sahel region ($0.711 \, \text{m}^2\text{g}^{-1}$ at 375 nm), which showcases the large variability on the absorption efficiency of dust depending on the source region. The backscatter fraction and the asymmetry parameter for dust events, 0.68 and 0.11, fall within the range of values found by Horvath et al. (2018) for Saharan dust, $0.71 \pm 0.03$ and $0.094 \pm 0.014$, respectively.



**Table 1.** Summary of mean values and standard errors for the measured and computed optical properties described in Sect. 2 during the full period, dust events, and during the three specific periods (A,B,C). Data are presented for selected wavelengths (grey-shaded rows) and weighted means (non-shaded rows). Results are categorized by inlet cut-off periods: PM2.5 (from 2019-09-04 00:00 UTC to 2019-09-26 12:00 UTC) and PM10 (from 2019-09-26 12:00 UTC to the end of the measurements). The number of measurements (N) corresponds to the 15 min averages used in the analysis.

| | units | **PM$_{2.5}$** | | | | | **PM$_{10}$** |
| --- | --- | --- | --- | --- | --- | --- | --- |
| | | **Full Period** (N=2156) | **Dust Events** (N=1160) | **Period A** (N=96) | **Period B** (N=96) | **Period C** (N=379) | **Full period** (N=464) |
| $b_{abs,370}$ | Mm$^{-1}$ | $3.87 \pm 0.48$ | $6.92 \pm 0.31$ | $5.11 \pm 0.52$ | $25.81 \pm 8.18$ | $2.10 \pm 0.10$ | $4.38 \pm 0.45$ |
| $b_{abs,880}$ | Mm$^{-1}$ | $0.66 \pm 0.05$ | $0.93 \pm 0.03$ | $0.72 \pm 0.06$ | $2.95 \pm 0.87$ | $0.49 \pm 0.01$ | $0.77 \pm 0.05$ |
| $b_{scat,525}$ | Mm$^{-1}$ | $83.72 \pm 12.25$ | $162.77 \pm 8.31$ | $114.16 \pm 12.18$ | $616.23 \pm 214.87$ | $40.70 \pm 2.23$ | $92.90 \pm 10.77$ |
| $b_{Bscat,525}$ | Mm$^{-1}$ | $9.09 \pm 1.15$ | $16.32 \pm 0.74$ | $12.39 \pm 1.14$ | $59.31 \pm 20.16$ | $4.91 \pm 0.21$ | $12.38 \pm 1.33$ |
| N | # | $43.14 \pm 2.45$ | $56.33 \pm 1.84$ | $47.97 \pm 3.49$ | $153.17 \pm 43.41$ | $34.93 \pm 0.65$ | $32.30 \pm 2.20$ |
| M | $\mu g\,m^{-3}$ | $26.38 \pm 3.28$ | $51.14 \pm 2.34$ | $37.24 \pm 4.16$ | $192.28 \pm 55.32$ | $13.09 \pm 0.76$ | $247.5 \pm 31.36$ |
| $R_{eff}$ | $\mu m$ | 0.469 | 0.545 | 0.528 | 0.579 | 0.417 | 1.649 |
| | | $0.445 \pm 0.004$ | $0.525 \pm 0.009$ | $0.517 \pm 0.003$ | $0.564 \pm 0.006$ | $0.420 \pm 0.004$ | $1.604 \pm 0.004$ |
| $MAE_{370}$ | $m^2\,g^{-1}$ | 0.156 | 0.140 | 0.141 | 0.139 | 0.178 | 0.019 |
| | | $0.180 \pm 0.003$ | $0.145 \pm 0.001$ | $0.148 \pm 0.003$ | $0.140 \pm 0.007$ | $0.191 \pm 0.004$ | $0.021 \pm 0$ |
| $MAE_{880}$ | $m^2\,g^{-1}$ | 0.041 | 0.020 | 0.021 | 0.016 | 0.058 | 0.004 |
| | | $0.048 \pm 0.001$ | $0.0233 \pm 0.0002$ | $0.024 \pm 0.001$ | $0.018 \pm 0.001$ | $0.051 \pm 0.002$ | $0.004 \pm 0$ |
| $MSE_{525}$ | $m^2\,g^{-1}$ | 3.41 | 3.67 | 3.19 | 3.73 | 3.25 | 0.396 |
| | | $3.30 \pm 0.034$ | $3.21 \pm 0.03$ | $3.25 \pm 0.06$ | $3.19 \pm 0.23$ | $3.31 \pm 0.04$ | $0.40 \pm 0.003$ |
| $MEE_{370}$ | $m^2\,g^{-1}$ | 3.66 | 3.73 | 3.29 | 3.66 | 3.90 | 0.391 |
| | | $3.98 \pm 0.052$ | $3.34 \pm 0.03$ | $3.40 \pm 0.06$ | $3.21 \pm 0.22$ | $4.19 \pm 0.062$ | $0.395 \pm 0.005$ |
| $MEE_{880}$ | $m^2\,g^{-1}$ | 3.61 | 3.88 | 3.39 | 4.18 | 3.12 | 0.469 |
| | | $2.94 \pm 0.040$ | $3.37 \pm 0.03$ | $3.38 \pm 0.06$ | $3.50 \pm 0.27$ | $2.78 \pm 0.04$ | $0.434 \pm 0.005$ |
| $SSA_{370}$ | – | 0.957 | 0.958 | 0.957 | 0.958 | 0.956 | 0.957 |
| | | $0.954 \pm 0.001$ | $0.955 \pm 0.0003$ | $0.956 \pm 0.001$ | $0.954 \pm 0.002$ | $0.954 \pm 0.001$ | $0.945 \pm 0.001$ |
| $SSA_{550}$ | – | 0.984 | 0.987 | 0.986 | 0.988 | 0.979 | 0.985 |
| | | $0.976 \pm 0.001$ | $0.985 \pm 0.0001$ | $0.985 \pm 0.001$ | $0.986 \pm 0.002$ | $0.972 \pm 0.001$ | $0.974 \pm 0.001$ |
| $SSA_{880}$ | – | 0.992 | 0.995 | 0.994 | 0.996 | 0.987 | 0.993 |
| | | $0.982 \pm 0.001$ | $0.993 \pm 0.0001$ | $0.993 \pm 0$ | $0.994 \pm 0$ | $0.978 \pm 0.001$ | $0.984 \pm 0.001$ |
| $AAE_{370-950}$ | – | 1.97 | 2.20 | 2.13 | 2.39 | 1.57 | 2.04 |
| | | $1.60 \pm 0.02$ | $2.02 \pm 0.01$ | $2.01 \pm 0.03$ | $2.23 \pm 0.04$ | $1.46 \pm 0.02$ | $1.70 \pm 0.01$ |
| $SAE_{450-635}$ | – | -0.010 | -0.100 | -0.077 | -0.183 | 0.214 | -0.092 |
| | | $0.31 \pm 0.019$ | $-0.05 \pm 0.005$ | $-0.038 \pm 0.01$ | $-0.131 \pm 0.021$ | $0.433 \pm 0.022$ | $-0.174 \pm 0.01$ |
| $SSAAE_{370-950}$ | – | -0.027 | -0.036 | -0.036 | -0.039 | -0.024 | -0.038 |
| | | $-0.027 \pm 0.001$ | $-0.036 \pm 0.003$ | $-0.036 \pm 0.001$ | $-0.039 \pm 0.002$ | $-0.024 \pm 0.00$ | $-0.055 \pm 0.002$ |
| $BF_{525}$ | – | 0.109 | 0.102 | 0.109 | 0.096 | 0.121 | 0.125 |
| | | $0.130 \pm 0.001$ | $0.112 \pm 0.0004$ | $0.117 \pm 0.001$ | $0.102 \pm 0.001$ | $0.135 \pm 0.001$ | $0.140 \pm 0.001$ |
| $g_{525}$ | – | 0.680 | 0.699 | 0.684 | 0.701 | 0.654 | 0.654 |
| | | $0.639 \pm 0.002$ | $0.679 \pm 0.001$ | $0.673 \pm 0.002$ | $0.694 \pm 0.002$ | $0.627 \pm 0.002$ | $0.631 \pm 0.001$ |
| $k_{525;sph}(\cdot 10^3)$ | – | – | 1.1 | 1.2 | 0.9 | – | – |
| | | – | $1.3 \pm 0.01$ | $1.32 \pm 0.04$ | $1.06 \pm 0.07$ | – | – |
| $k_{525;triax}(\cdot 10^3)$ | – | – | 1.7 | 1.9 | 1.6 | – | – |
| | | – | $1.89 \pm 0.01$ | $1.94 \pm 0.06$ | $1.63 \pm 0.10$ | – | – |





The higher absorption efficiency in Period C associated to the presence of anthropogenic aerosol particles, happened espe-
cially between midnight and noon, as denoted by the AAE and SAE close to 1, a close to zero or positive SSAAE, a significantly
smaller effective radius (0.42 vs 0.54 μm), and a smaller SSA at 880 nm (around 0.978) showcasing both the larger relative
contribution of fine particles (González-Flórez et al., 2023) and the higher imaginary refractive index ($k$). Related with that,
the mass absorption efficiency strongly increases from $\sim 0.145$ to $\sim 0.191$.

Although the average of the entire measurement period exhibits a clear optical dust signature, the overall values are in-
fluenced by those observed during Period C, highlighting that even in remote and relatively isolated dust-source locations,
regional and local anthropogenic pollution can impact dust measurements.

The last row of Fig. 2 illustrates the temporal evolution of the estimated imaginary part of the complex refractive index ($k$)
at 525 nm for different shape assumptions: spheres (solid line), triaxial ellipsoids (dashed line), and the full range of values
considered in Sect. 2.6. During dust events, $k$ remains relatively stable, while Period C and the calm phases of Periods A and B
exhibit increases of up to an order of magnitude, highlighting a stronger influence of absorbing anthropogenic particles during
these periods.

## 3.2 Optical properties and their relationship with effective radius

At this measurement site, mineral dust fluxes, wind friction velocity, and particle size are closely interrelated (González-Flórez
et al., 2023). In this section, we examine the dependence of the observed optical properties on the effective radius (R$_{\text{eff}}$, cf.
Section 2.3). It is important to note that the campaign consisted of two distinct measurement periods with different inlet cut-offs
(PM2.5 and PM10), which strongly influence R$_{\text{eff}}$ due to the dominance of coarse particles during dust emission events.

Figure 3 features the absorption ($b_{abs}$) and scattering ($b_{scat}$) coefficients, and SSA as a function of R$_{\text{eff}}$. The upper panels
correspond to the PM2.5 period, while the lower panels represent the PM10 period. The average absorption Ångström expo-
nent (AAE) for each R$_{\text{eff}}$ bin is also indicated in Figs. 3a,d. For the PM2.5 period, $b_{abs}$ exhibits a non-monotonic U-shaped
dependence on R$_{\text{eff}}$, initially decreasing until the 0.4–0.5 μm bin before increasing again at larger sizes. In contrast, $b_{scat}$
increases consistently for R$_{\text{eff}} > 0.4$ μm. The SSA also shows a clear dependence on R$_{\text{eff}}$, with distinct spectral variations
between the PM2.5 and PM10 periods. The observed U-shaped trend in $b_{abs}$ reflects different source contributions during the
campaign: smaller R$_{\text{eff}}$ values (AAE $\sim$1) correspond to periods influenced by transported fine carbonaceous particles, while
larger R$_{\text{eff}}$ values are associated with local and nearby dust emissions. At smaller R$_{\text{eff}}$, $b_{abs}$ decreases due to increasing SSA,
despite similar $b_{scat}$ values across the first three size bins. For larger R$_{\text{eff}}$, SSA stabilizes, and $b_{scat}$ and $b_{abs}$ increase with
dust abundance. The spectral dependence of SSA during the PM2.5 period aligns with a mix of fine carbonaceous aerosols and
coarser dust particles. The PM10 period exhibits optical behavior consistent with the larger R$_{\text{eff}}$ bins of the PM2.5 period.

The spectral dependence of b$_{\text{abs}}$, b$_{\text{scat}}$, and SSA is further examined through their respective Ångström exponents (Fig. 4).
The AAE increases with effective radius, particularly in the PM2.5 period for R$_{\text{eff}}$ values above 0.45 μm, where it rises from
approximately 1.2 to 2.5. This trend is consistent with increasing friction velocity ($u_*$), indicating stronger local dust emissions.
Indeed, Fig. S7 shows that the increase in R$_{\text{eff}}$ during the PM2.5 period with wind the friction velocity is also associated to



**Figure 2.** Time evolution of key optical and physical properties during the highlighted measurement periods: a) Period A (midnight 21st to morning 22nd September 2019), b) Period B (midnight 6th to morning 7th September 2019), and c) Period C (midnight 7th to noon 11th September 2019). Each row represents the time series of: absorption ($b_{abs}$) and scattering ($b_{scat}$) coefficients at 370 and 880 nm; mass concentration ($M_{PM}$) and effective radius ($M_{eff}$); mass absorption, scattering, and extinction efficiency (MAE, MSE, MEE) at 370 and 880 nm; single scattering albedo (SSA) at 370 and 880 nm; absorption, scattering, and SSA Ångström exponents (AAE, SAE, SSAAE); and the estimated imaginary part of the refractive index ($k$) at 525 nm for spheres (solid line), triaxial ellipsoids (dashed line), and a range of values for other shape assumptions described in Sect. 2.6





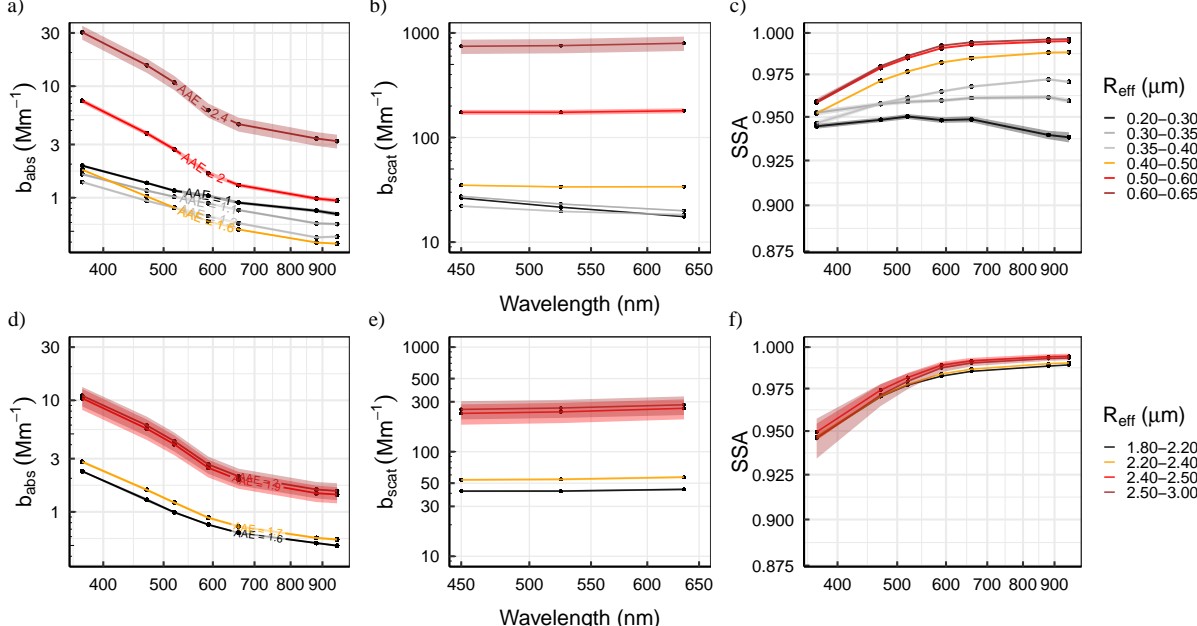

**Figure 3.** Mean and standard error (se) values of the absorption (a,d) and scattering (b,e) coefficients, and the single scattering albedo (SSA; c,f) as a function of wavelength (x-axis) and effective radius (colour scale) at M'Hammid for both PM2.5 (upper panel) and PM10 (lower panel) periods. The log-log plot highlights the enhanced absorption efficiency of dust at short-UV wavelengths, where absorption increases more significantly compared to longer wavelengths. In contrast, BC-influenced particles typically exhibit a nearly linear spectral dependence. The absorption Ångström exponent (AAE) values are indicated along each absorption coefficient curve.

higher AAE measurements. In contrast, during the PM10 period, the AAE exhibits a more gradual and steady increase with $R_{eff}$ and $u_*$.

Figure 4 also shows a strong relationship between SAE and effective radius. During the PM2.5 period, SAE decreases

significantly, from around 1 to values near or below 0, as $R_{eff}$ increases. During the PM10 period, the high concentration of coarse particles results in SAE remaining consistently low, below 0. A negative SSAAE has been recognized as a key indicator of Saharan dust events over Europe (Collaud Coen et al., 2004), and this criterion is met for $R_{eff}$ values above 0.3 µm during the PM2.5 period and throughout the entire PM10 period.

Overall, these results indicate that during the PM2.5 period, where fine particles were sampled, the influence of dust particles

on optical properties was only prominent under strong emission conditions. In contrast, during the PM10 period, the presence of coarser particles led to a consistently stronger dust optical signature, significantly shaping the spectral behavior of $b_{abs}$, $b_{scat}$, and SSA.

Additionally, we computed two key parameters related to the scattering phase function of the particles: the backscatter fraction (BF) and the asymmetry parameter ($g$). As shown in Fig. 5, forward scattering (indicated by higher values of $g$ and




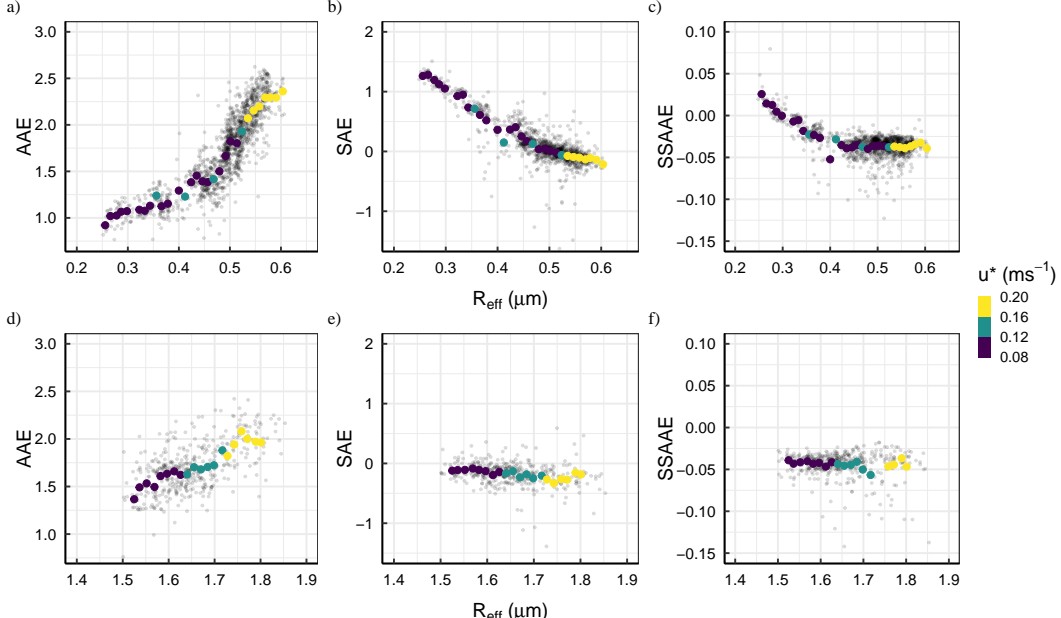

**Figure 4.** Scatter plot of the Absorption Ångström Exponent, AAE (a,d), the Scattering Ångström Exponent, SAE (b,e), and the single-scattering albedo Ångström exponent (SSAAE) as a function of the particle effective radius ($R_{eff}$) —x-axis— at M'Hammid for both PM2.5 (upper panel) and PM10 (lower panel) periods. The colour scale indicates the average friction velocity, $u_*$, for each $R_{eff}$ bin, with this averaging resulting in the smaller range of $u_*$ values. Bins were determined using the Freedman-Diaconis rule (Freedman and Diaconis, 1981) to ensure an optimal balance between resolution and statistical robustness.

lower values of BF) increases with particle effective radius ($R_{eff}$). The spectral dependency of these parameters shifts as particles become coarser, with the slope of BF transitioning from negative to positive and that of $g$ from positive to negative. This trend indicates that finer particles exhibit stronger backscattering at shorter wavelengths, while coarser particles favour forward scattering. During the PM2.5 period, for $R_{eff} > 0.4 \mu m$, the BF($g$) decreases(increases) with the wavelength from $\sim 0.1(0.68)$ at the short-UV to $\sim 0.09(0.7)$ at the near-infrared. During the PM10 period, these spectral variations are less

pronounced, with BF values ranging between 0.12 and 0.16 and $g$ values between 0.6 and 0.64. These values align with previously reported BF and asymmetry parameters. Horvath et al. (2018) found for Saharan mineral dust advected to the Western Mediterranean a $g$ ranging from 0.75 to 0.65 and a BF ranging from 0.07 to 0.12, respectively, which matches the results found here in the source area, specially in the case of PM2.5. Similarly, Ryder et al. (2018) found a $g$ of 0.74 for airborne measurements within the Saharan Aerosol Layer over the Atlantic.

As shown in Figure 6, regardless of the inlet cut-off period (PM2.5 or PM10), MAE consistently decreases as particle size increases. For the PM2.5 period (noting the logarithmic y-axis), MAE values drop by approximately an order of magnitude, from $\sim 0.30$ to $0.15$ $m^2 g^{-1}$ at short-UV wavelengths and from $\sim 0.12$ to $0.02$ $m^2 g^{-1}$ at near-infrared wavelengths.





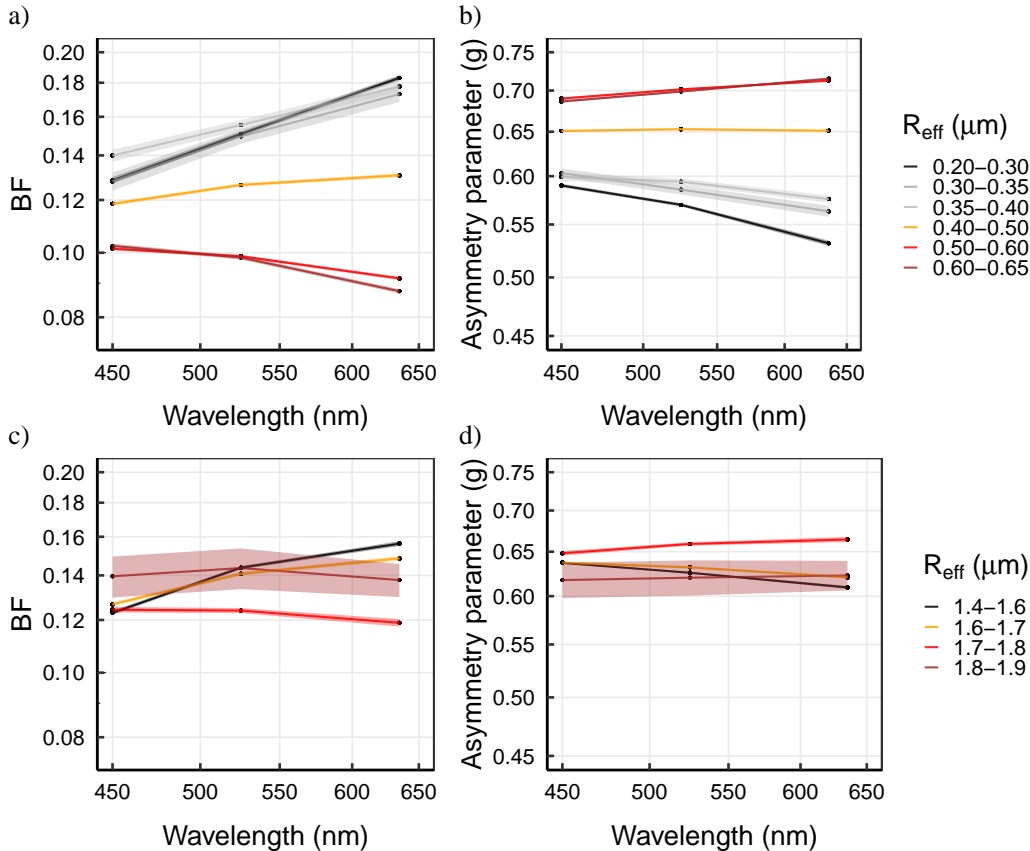

**Figure 5.** Mean (lines) and standard error (se, shaded colours) values of the backscatter fraction, BF (a,c) and the asymmetry parameter (b,d) as a function of the wavelength (x-axis in logarithmic scale) and the effective radius (colour scale) at M'Hammid for both PM2.5 (upper panel) and PM10 (lower panel) periods.

Conversely, the scattering, and consequently the extinction efficiency, presented a different pattern across the two inlet cut-offs periods. During the PM2.5 period, MSE decreases with wavelength for particles with $R_{eff}$ below 0.4 μm, consistent with the stronger scattering efficiency of fine particles at shorter wavelengths (Hansen and Travis, 1974). For coarser particles, MSE remains similar. In the PM10 period, MSE declines significantly, both compared to the PM2.5 period and as $R_{eff}$ increases. This trend reflects the shifting peak scattering efficiency toward longer wavelengths, beyond the measurement range of this study (Hansen and Travis, 1974). The cumulative effect of high absorption and scattering efficiencies, particularly for smaller particles likely dominated by fine carbonaceous aerosols, is captured in the MEE trends (Figs. 6c,d). These trends highlight the distinct optical behaviours of different aerosol sources, with fine particles exhibiting higher MAE and MSE, while coarser dust particles contributing more strongly to total scattering.





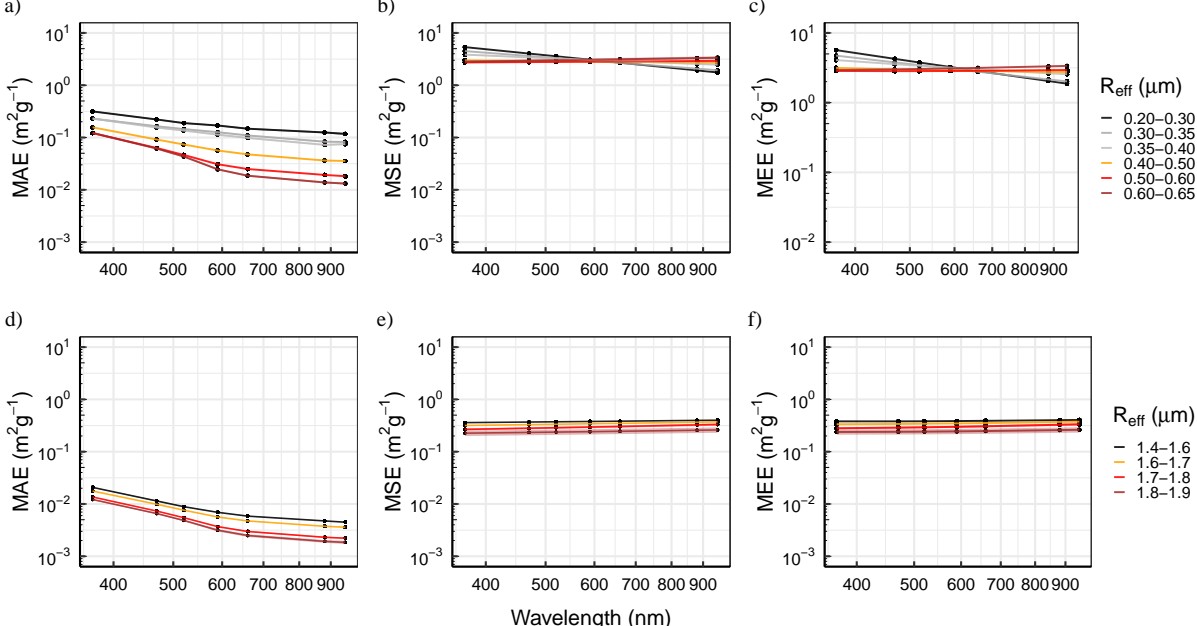

**Figure 6.** Mean and standard error (se) values of the mass absorption, MAE (a,d), scattering, MSE (b,e), and extinction, MSE (c,f) efficiency as a function of the wavelength (x-axis in logarithmic scale( and the effective radius (colour scale) at M'Hammid for both PM2.5 (upper panel) and PM10 (lower panel) periods.

### 3.3 Imaginary part of the refractive index of mineral dust

This section presents the retrieval of the imaginary part of the complex refractive index ($k$) for mineral dust, following the methodology described in Sect. 2.6. The analysis focuses exclusively on periods classified as dust events (see Sect. 3.1) during the PM2.5 measurement period. For each of the 15-minute averaged measurements, the value of $k$ was retrieved (see for example the last row of Fig. 2), these $k$s are referred to as the average (of $k$) of the retrievals. Complementarily, $k$ retrievals over the measurements flagged as dust events in Sect. 3.1 were also computed by taking as input the time averaged measurements of scattering, absorption and PSD, which lead to a set of single estimates of $k$, which we refer to through the text as the retrieval (of $k$) of the average.

Fig. 7 summarizes the results, illustrating the spectral dependence of $k$ under different particle shape assumptions (Fig. 7a). Panels b-d show the variability in $k$ due to different values of the real part of the refractive index ($n$, from 1.48 to 1.55 in steps of 0.01) and the three inlet cut-off diameter efficiencies (1.7 and 1.7 $\pm$ 10%) for spheres, biaxial ellipsoids and triaxial ellipsoids. For comparison, results for the Moroccan sample from Di Biagio et al. (2019) (dB19) are also included.

All panels show an overall decrease of $k$ with increasing wavelength across the five shortest wavelengths. The dB19 values of $k$ in Fig.7a are similar to our retrievals at shorter wavelengths but fall toward the lower end of our $k$ estimates at longer wavelengths. The slope of the dB19 $k$ curve in the shorter wavelengths is steeper than in our retrievals. Additionally, our





retrievals assuming spherical particles are consistently lower than the dB19 values. Although Di Biagio et al. (2019) also assumes spherical particles, several factors may explain the discrepancies between their results and ours. First, the samples are not identical; while both originate from nearby regions on a global scale, the dust in our field campaign may not be directly comparable to the resuspended dust used in dB19. Second, we employed a PM2.5 inlet, whereas Di Biagio et al. (2019) used a PM10 inlet. This difference can influence the retrievals if the composition and shape of the particles changes vary with size (Panta et al., 2023), even though the retrievals assume size-independent $k$. Third, different inlets for measuring the PSD and the optical properties, which introduced additional uncertainties in our retrievals (see Table 2), likely more strongly affecting the longer wavelengths. Another contributing factor is the potential influence of anthropogenic aerosols in our samples (Sect. 3.1), which may enhance overall absorption and lead to higher $k$ values. While this effect is important when the average of $k$ is used as comparison, it is less relevant when $k$ is computed from the averaged measurements (Fig. S8). Finally, our retrieval procedure includes consistent corrections to the Fidas PSD for particle shape and refractive index, which is not the case in dB19, although the impact of this difference may be relatively small.

Panels b,c and d of Fig. 7 feature the distribution of 15-min $k$ retrievals for three particle shape assumptions. In each case, the average of these retrievals (indicated by dots) is consistently equal to or higher than the $k$ value derived from the averaged optical measurements (indicated by triangles). We attribute this systematic difference to the over-representation of absorbing aerosols other than dust (cf. Sect. 3.1) in the time averaged data. This occurs because periods with low dust concentrations —potentially richer in more absorbing particles— are weighted equally with high-dust events in the averaging process.

The violin plots in Fig. 7 illustrate the spread of $k$ values obtained under eight different assumptions for the real part of the refractive index ($n$), across three simulated cut-off of the inlet. The contributions of cut-off diameter and $n$ to the variability in $k$ are further detailed in Tables 2 and 3. Table 2 shows the time average estimate of $k$ presents time-averaged $k$ values at selected wavelengths for the three cut-off diameters, averaged over all $n$ values. On the other hand, Table 3 reports $k$ estimates as a function of $n$, averaged over the different inlet cut-off values. Both tables include results for a subset of particle shapes; the complete wavelength range is available in Tables S1–S4.

The results in Table 2 reveal a strong dependence of $k$ on the assumed inlet cut-off diameter. Specifically, retrievals yield lower $k$ values when larger particles are included, and higher values when the cut-off diameter is reduced. This dependency is asymmetric: increasing the cut-off diameter by 10% results in a decrease in $k$ of approximately 6–15%, whereas decreasing it by 10% leads to an increase of about 9–19% relative to the 1.7 µm baseline.

A similar, though weaker, sensitivity is observed for the assumed value of $n$. Table 3 shows that $k$ decreases slightly with increasing $n$, with maximum $k$ values occurring at the lower end of the $n$ range. For example, varying $n$ from 1.48 to 1.55 (a 5% range around a central estimate of 1.49) causes changes in $k$ on the order of 0.0002 at 370 nm for irregular particles. Extrapolating this to a 10% change in $n$ suggests a maximum sensitivity of around 14%, which—although modest—is comparable to the sensitivity to inlet cut-off diameter. However, a 10% uncertainty in $n$ is unrealistic, as reported values for mineral dust typically fall within a much narrower range; for instance, Di Biagio et al. (2019) reports values between 1.48 and 1.55, a variation of less than 5% from the lower bound.




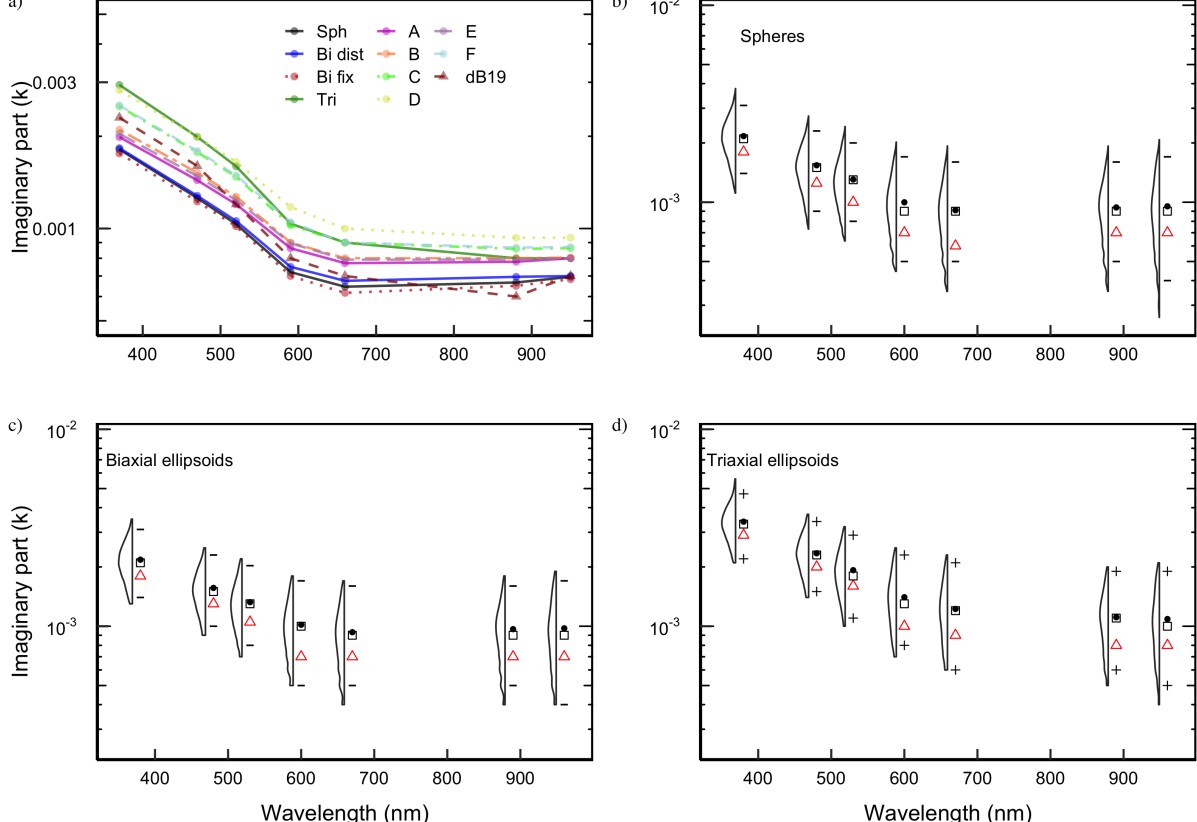

**Figure 7.** Imaginary part of the complex refractive index ($k$) as a function of wavelength. (a) Coloured lines indicate the average $k$ retrieved under different shape assumptions, spheres, bi-axial, tri-axial and irregular shapes –A to F from Gasteiger and Wiegner (2018)– using the best-estimate real refractive index (1.49) and an inlet cut-off of 1.70 µm, as well as the $k$ from Di Biagio et al. (2019) for the Morocco sample as dB19; (b–d) Distributions of $k$ for spheres (b), biaxial ellipsoids (c), and triaxial ellipsoids (d) during dust events, showing the mean (dot), median (square), mean of the retrievals of the average (red triangle), and 5th to 95th percentiles (flat segments). Each distribution reflects the range of retrievals for various real refractive index values and inlet cut-off assumptions, highlighting the sensitivity of $k$ to shape and measurement parameters.

The third factor explored in Tables 2 and 3 is particle shape. The variability in $k$ due to shape assumptions is larger than that from either $n$ or inlet cut-off diameter. Depending on wavelength and configuration, differences between particle shapes can account for 20–54% variation in the median $k$ value.

Spheres and biaxial ellipsoids show similar $k$ estimates, whereas triaxial ellipsoids and shapes C, D, and F produce signifi-
cantly higher values (Fig. 7a). This outcome is consistent with the findings of Kong et al. (2024), who demonstrated through numerical experiments that greater particle non-sphericity leads to higher retrieved $k$ values. To verify this in our study, we used the ratio of volume-equivalent to cross section equivalent particle radii ($\xi_{vc}$) from Gasteiger et al. (2011). Shapes C, D,





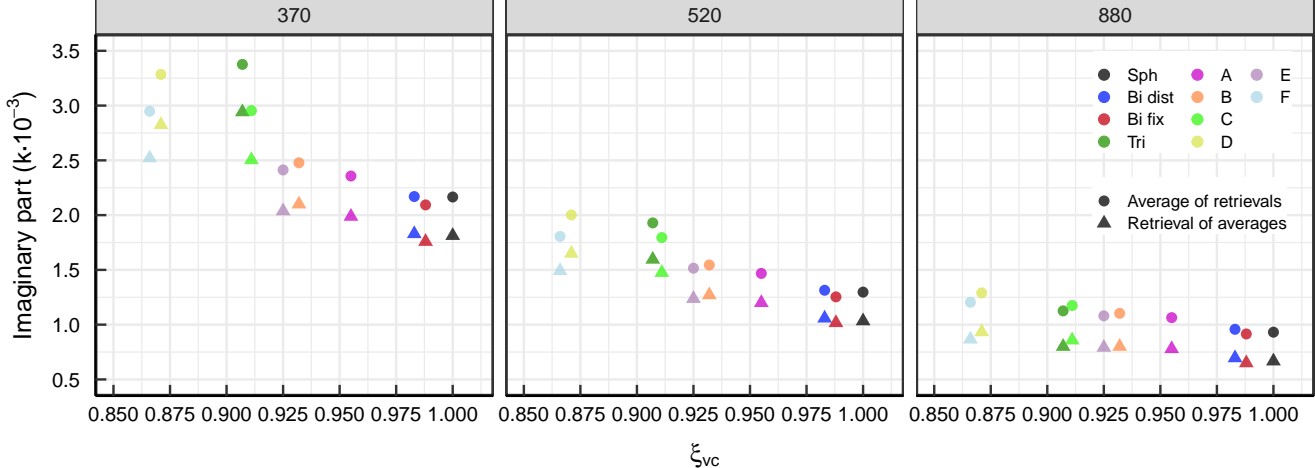

**Figure 8.** Imaginary part of the complex refractive index ($k$) retrieved for the same shape assumptions as in Fig. 7 as a function of the asphericity of the particles as defined by Gasteiger et al. (2011), $\xi_{vc}$ for three wavelengths (370, 520 and 880 nm) for the retrieval of $k$ for the averaged optical values (triangles) and for the temporal average of the retrievals filtered for dust events (points).

and F and triaxial ellipsoids feature lower $\xi_{vc}$ values (0.911 , 0.871, 0.866 and 0.907, respectively) than irregular shapes A, B and E, biaxial ellipsoids and their distribution (0.955, 0.932, 0.925, 0.983 and 0.988, respectively), confirming that more

elongated or irregular particles tend to produce higher $k$ retrievals than spherical particles.

As a results, $k$ estimated retrieved using triaxial ellipsoids can be up to 60% higher than those from spherical assumptions at shorter wavelengths, and around 30% higher at longer wavelengths. This relationship is further illustrated in Fig. 8, which shows the correlation between retrieved $k$ values and particle shape, as quantified by $\xi_{vc}$. A relatively linear trend is observed across all three panels, with increasing $k$ values corresponding to more non-spherical particles (i.e., lower $\xi_{vc}$). The slope of

the linear fit between $\xi_{vc}$ and $k$ becomes less negative as the wavelength increases. While this relationship appears consistent, it is based on a limited number of data points, and any extrapolation should therefore be treated with caution. Nonetheless, establishing a robust connection between particle non-sphericity and retrieved $k$ could provide a valuable tool for reinterpreting past measurements and datasets—many of which rely on spherical particle assumptions—within the context of more realistic, non-spherical dust models.

### 3.3.1 Comparison with dust-dominated AERONET retrievals

Figure 9 shows the comparison of our in-situ $k$ estimates —extrapolated to the AERONET wavelengths of 440, 550, 675, 870, and 1020 nm— and AERONET dust-filtered retrievals compiled over two decades from various mid-latitude dust emission regions. These include Ouarzazate (OUARZ), western and eastern North Africa (WNAFR, ENAFR), the Sahel (SAHEL), the Arabian Peninsula (ARPEN), central Asia (CASIA), and India (INDIA). Our retrievals are shown for three particle shape

assumptions: spheres, biaxial ellipsoids, and triaxial ellipsoids.





**Table 2.** Imaginary part of the complex refractive index $k$ for periods characterized as *dust events* throughout the PM2.5 part of the campaign. The retrieval of the averaged optical measurements is shown in bold, while the 5th percentile, the mean, and the 95th percentile of the distribution from the 15-min retrievals are shown within the parenthesis. All retrievals in this table were computed under the assumption of $n = 1.49$ and they are shown multiplied by $10^3$ for clarity. They have been obtained for different shapes: spheres, distribution of biaxial ellipsoids, the average of the A to F irregular shapes from Gasteiger and Wiegner (2018), and the triaxial ellipsoid shape; for 3 wavelengths ranging between the near-UV to the near-infrared and 3 different inlet efficiency cut-off assumptions.

| Shape | $\lambda$ (nm) | $\phi_{cut}$=1.53 $\mu m$ | $\phi_{cut}$=1.70 $\mu m$ | $\phi_{cut}$=1.87 $\mu m$ |
|---|---|---|---|---|
| **Spheres** | 370 | **2.1** (1.6-2.4-3.7) | **1.8** (1.3-2.1-3.0) | **1.6** (1.2-1.9-2.7) |
| | 520 | **1.2** (0.9-1.4-2.3) | **1.1** (0.8-1.2-2.0) | **0.9** (0.7-1.1-1.8) |
| | 880 | **0.8** (0.5-1.0-1.8) | **0.7** (0.5-0.9-1.6) | **0.6** (0.4-0.8-1.4) |
| **Biaxial** | 370 | **2.1** (1.5-2.5-3.4) | **1.8** (1.4-2.1-3.1) | **1.6** (1.2-1.9-2.9) |
| | 520 | **1.2** (0.9-1.4-2.4) | **1.1** (0.8-1.3-2.1) | **1.0** (0.7-1.1-1.8) |
| | 880 | **0.8** (0.5-1.0-1.8) | **0.7** (0.5-0.9-1.6) | **0.6** (0.4-0.8-1.5) |
| **Triaxial** | 370 | **3.4** (2.6-3.7-5.8) | **3.0** (2.2-3.4-4.6) | **2.6** (2.0-3.0-4.1) |
| | 520 | **1.8** (1.4-2.1-3.3) | **1.6** (1.2-1.8-2.9) | **1.4** (1.0-1.6-2.6) |
| | 880 | **0.9** (0.6-1.2-2.1) | **0.8** (0.6-1.1-1.9) | **0.7** (0.5-1.0-1.7) |
| **Irregular** | 370 | **2.7** (1.9-3.0-4.6) | **2.4** (1.7-2.7-4.0) | **2.1** (1.6-2.4-3.7) |
| | 520 | **1.6** (1.1-1.8-3.0) | **1.4** (1.0-1.6-2.7) | **1.3** (0.9-1.5-2.4) |
| | 880 | **0.9** (0.6-1.2-2.1) | **0.8** (0.6-1.1-1.9) | **0.8** (0.5-1.0-1.8) |

**Table 3.** Imaginary part of the complex refractive index $k$, with the same table structure as in Table 2 but fixing the best estimate of the inlet efficiency cut-off to $\phi_{cut} = 1.70$ and varying the $n$ between 8 different values. Please note the in the table $k$ is multiplied by $10^3$.

| Shape | $\lambda$ (nm) | n=1.48 | n=1.49 | n=1.50 | n=1.51 | n=1.52 | n=1.53 | n=1.54 | n=1.55 |
|---|---|---|---|---|---|---|---|---|---|
| **Spheres** | 370 | **1.8** (1.3-2.1-3.0) | **1.8** (1.3-2.1-3.0) | **1.8** (1.4-2.1-3.0) | **1.8** (1.3-2.1-2.9) | **1.8** (1.3-2.1-2.9) | **1.8** (1.3-2.1-3.2) | **1.8** (1.3-2.1-3.2) | **1.8** (1.3-2.1-2.9) |
| | 520 | **1.1** (0.8-1.3-2.0) | **1.1** (0.8-1.2-2.0) | **1.0** (0.8-1.2-2.0) | **1.0** (0.8-1.2-2.0) | **1.0** (0.8-1.2-2.0) | **1.0** (0.7-1.2-2.0) | **1.0** (0.7-1.2-2.0) | **1.0** (0.7-1.2-2.0) |
| | 880 | **0.7** (0.5-0.9-1.6) | **0.7** (0.5-0.9-1.6) | **0.7** (0.5-0.9-1.6) | **0.7** (0.5-0.9-1.5) | **0.7** (0.5-0.9-1.5) | **0.6** (0.4-0.9-1.6) | **0.6** (0.4-0.9-1.6) | **0.6** (0.4-0.9-1.6) |
| **Biaxial** | 370 | **1.9** (1.4-2.1-3.1) | **1.8** (1.4-2.1-3.1) | **1.8** (1.4-2.1-3.3) | **1.8** (1.4-2.1-3.0) | **1.8** (1.3-2.1-3.0) | **1.8** (1.3-2.1-2.9) | **1.8** (1.3-2.1-2.9) | **1.8** (1.3-2.1-2.9) |
| | 520 | **1.1** (0.8-1.3-2.1) | **1.1** (0.8-1.3-2.1) | **1.1** (0.8-1.3-2.0) | **1.1** (0.8-1.2-2.0) | **1.0** (0.8-1.2-2.0) | **1.0** (0.8-1.2-2.0) | **1.0** (0.8-1.2-2.0) | **1.0** (0.8-1.2-2.0) |
| | 880 | **0.7** (0.5-0.9-1.6) | **0.7** (0.5-0.9-1.6) | **0.7** (0.5-0.9-1.6) | **0.7** (0.5-0.9-1.6) | **0.7** (0.5-0.9-1.6) | **0.7** (0.5-0.9-1.6) | **0.7** (0.5-0.9-1.6) | **0.7** (0.5-0.9-1.6) |
| **Triaxial** | 370 | **3.0** (2.2-3.4-4.6) | **3.0** (2.2-3.4-4.6) | **3.0** (2.2-3.3-4.6) | **2.9** (2.1-3.3-4.5) | **2.9** (2.1-3.3-4.5) | **2.9** (2.2-3.3-4.8) | **2.9** (2.1-3.3-4.5) | **2.8** (2.1-3.3-4.5) |
| | 520 | **1.6** (1.2-1.8-2.9) | **1.6** (1.2-1.8-2.9) | **1.6** (1.2-1.8-2.9) | **1.6** (1.2-1.8-2.9) | **1.6** (1.2-1.8-2.9) | **1.6** (1.1-1.9-2.9) | **1.6** (1.1-1.8-2.9) | **1.5** (1.1-1.8-2.9) |
| | 880 | **0.8** (0.6-1.1-1.9) | **0.8** (0.6-1.1-1.9) | **0.8** (0.6-1.1-1.9) | **0.8** (0.6-1.1-1.9) | **0.8** (0.6-1.1-1.9) | **0.8** (0.6-1.1-1.9) | **0.8** (0.6-1.1-1.9) | **0.8** (0.6-1.1-1.9) |
| **Irregular** | 370 | **2.4** (1.7-2.7-4.2) | **2.4** (1.7-2.7-4.0) | **2.3** (1.7-2.7-4.1) | **2.3** (1.7-2.6-4.0) | **2.3** (1.7-2.6-4.0) | **2.3** (1.7-2.6-4.0) | **2.2** (1.6-2.6-4.0) | **2.2** (1.6-2.5-3.9) |
| | 520 | **1.4** (1.0-1.6-2.7) | **1.4** (1.0-1.6-2.7) | **1.4** (1.0-1.6-2.7) | **1.4** (1.0-1.6-2.6) | **1.4** (1.0-1.6-2.6) | **1.4** (1.0-1.6-2.6) | **1.3** (0.9-1.6-2.6) | **1.3** (0.9-1.6-2.5) |
| | 880 | **0.8** (0.6-1.1-1.9) | **0.8** (0.6-1.1-1.9) | **0.8** (0.6-1.1-2.0) | **0.8** (0.6-1.1-1.9) | **0.8** (0.6-1.1-1.9) | **0.8** (0.6-1.1-1.9) | **0.8** (0.6-1.1-2.0) | **0.8** (0.6-1.1-2.0) |

Due to differences in spatio-temporal representativeness and observational context, a strict quantitative comparison between AERONET and in-situ $k$ values is not appropriate. Nevertheless, we observe qualitative agreement with the AERONET retrievals from OUARZ, the nearest station, located approximately 200 km north of our measurement site.



The strongest alignment occurs under the triaxial shape assumption, where median $k$ values closely match those from AERONET at 550 and 675 nm, and the full distribution shows substantial overlap at longer wavelengths. At 440 nm, however, our in-situ estimates diverge more noticeably from the AERONET retrievals. When assuming spherical or biaxial shapes, our $k$ values are generally lower than both the triaxial-based retrievals and the AERONET values from OUARZ, especially at shorter wavelengths (440–550 nm).

A key observation from Fig. 9 is the pronounced regional variability in $k$. While values retrieved at OUARZ closely align with our in-situ measurements, the median $k$ increases progressively from western to eastern North Africa and further across the Sahel, reaching even higher values in central Asia and India. These differences are likely driven by regional variations in dust mineralogy, particularly differences in iron oxide content that influence absorption properties.

Part of this variability may also be affected by residual contamination from non-dust aerosols in AERONET retrievals, despite the rigorous filtering applied. Additionally, inherent uncertainties in AERONET retrievals contribute to these differences, including potential biases arising from its limited sensitivity to super-coarse dust particles, which could influence both the retrieved particle size distribution and associated $k$ values. Moreovover, assumptions regarding particle sphericity in AERONET inversions may also impact the accuracy of retrieved absorption properties Adebiyi et al. (2023).

These results emphasize the critical need for region-specific in-situ studies, such as the present work, to improve constraints on dust optical properties. Direct measurements of locally emitted dust help refine our understanding of regional differences and provide essential data for improving global climate models, particularly in quantifying dust absorption and its role in the Earth's radiative budget.

## 4 Summary and conclusions

This study provides new high-resolution, source-region in-situ measurements of mineral dust optical properties, with a specific focus on absorption, scattering, and their spectral dependencies. The measurements, conducted in a dust source of the Moroccan Sahara, offer crucial insights into the morphological and optical characteristics of freshly emitted dust and their implications for climate modeling and remote sensing retrievals.

The campaign captured three distinct aerosol scenarios: local dust emissions, haboob-driven dust events, and anthropogenic pollution advection mixed with minor dust emissions. Dust-dominated periods exhibited high SSA values, increasing from 0.955 at 370 nm to 0.985 at 550 nm to 0.993 at 880 nm for PM2.5 particles and from 0.945 at 370 nm to 0.974 at 550 nm to 0.985 at 880 nm for PM10 particles, consistent with previous Saharan dust observations. Periods with anthropogenic influence showed lower SSA, particularly at near-infrared wavelengths and higher mass absorption efficiencies, suggesting the presence of fine carbonaceous aerosols.

SSA, scattering, and absorption coefficients were shown to be strongly dependent on the particle effective radius ($R_{eff}$). The AAE increased with $R_{eff}$, reaching values above 2 during intense dust events, indicative of enhanced short-wave absorption by iron oxides. The SAE decreased to negative values for larger dust particles, further confirming the dominance of coarse-mode mineral dust during emission events.





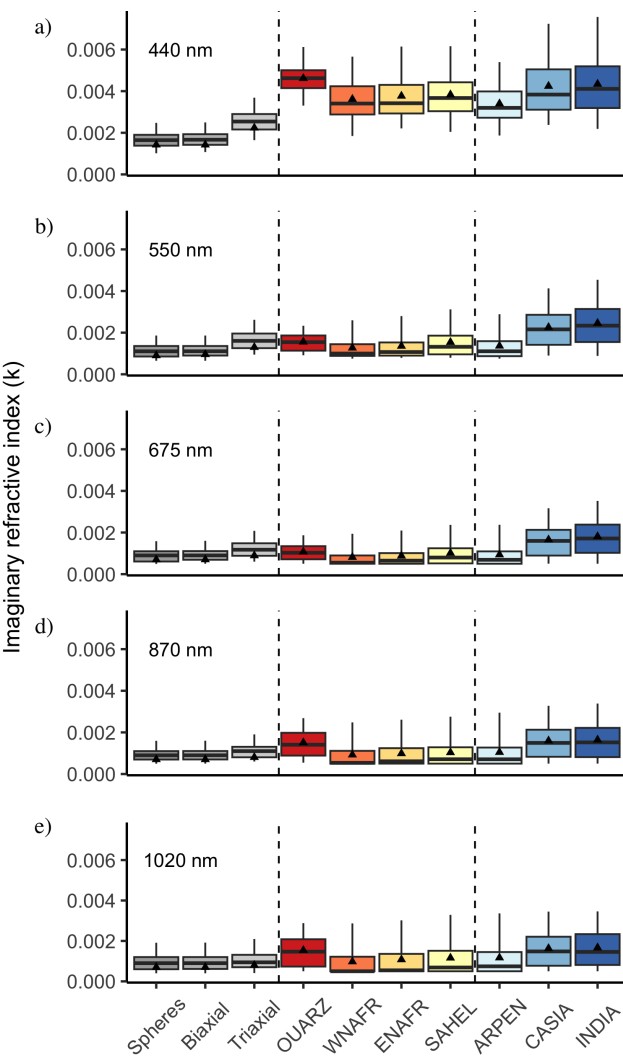

**Figure 9.** Box-plots of the imaginary part of the complex refractive index obtained for the dust events as described in Sec. 2.6 for the assumption of the best estimates of $n$ and inlet cut-off for spheres, and biaxial and triaxial ellipsoids (in greyscale), and from 20 years of AERONET measurements over different mid-latitude dust emitting areas (from closer to further from our location, in colours). The retrievals of the average for the in-situ measurements and the average for the AERONET estimations are also shown through the black triangles. The regions of the AERONET measurements are shown in Fig S9, with the following acronyms: Ouarzazate (OUARZ), northwestern Africa (WNAFR), northeastern Africa (ENAFR), Sahel (SAHEL), Arabian Peninsula (ARPEN), central Asia (CASIA), and India (INDIA). The box-plots limits represent the 25th and 75th quantiles and the lines represent the 5th and 95th quantiles.

Forward scattering (asymmetry parameter, $g$) increased with $R_{eff}$, while BF decreased, highlighting the role of particle size in modifying dust scattering behavior. The MAE decreased significantly with increasing $R_{eff}$, for instance from $\sim 0.30$



to $0.15\text{m}^2\text{g}^{-1}$ at 370 nm, reflecting reduced absorption efficiency for larger dust particles. The MSE displayed a distinct
wavelength dependency, with stronger scattering for fine particles and a shift toward longer wavelengths for coarse particles.

We have retrieved values of the imaginary part of the complex refractive index ($k$) for the campaign measurements, with aspect ratio constraints derived from single particle analysis of samples collected during the campaign. The retrieved $k$ are comparable to those obtained by Di Biagio et al. (2019) in a resuspension chamber from a Moroccan sediment sample. Our central estimates under spherical and biaxial ellipsoid assumptions yielded lower values (e.g., $k \sim 0.0011$ at 520nm), while
assuming triaxial or more irregular particle shapes resulted in higher values (e.g., $k \sim 0.0016$ at 520nm). This trend —an increase in retrieved $k$ with greater particle non-sphericity— highlights the sensitivity of absorption retrievals to morphological assumptions. It also underscores the need to account for realistic particle geometries to reduce uncertainty in estimating dust absorption properties from in situ measurements.

Comparisons with AERONET-retrieved $k$ values show general agreement, particularly in the near-infrared. However, dis-
crepancies appear at shorter wavelengths. These differences likely stem from a combination of factors, including methodological limitations in both measurements and retrievals, the intrinsic mismatch between column-integrated climatological data (AERONET) and point-based campaign observations, and the potential influence of externally mixed absorbing species (e.g., black carbon), which may persist in AERONET retrievals despite the application of dust filters.

Finally, we found that modelling the effects of the measurement inlet cut-off significantly influences the retrievals, empha-
sizing the importance of instrument design considerations when interpreting optical properties of mineral dust.

The findings of this study improve our understanding of freshly emitted mineral dust optical properties and their dependence on size, shape, and composition. These results contribute to refining dust parametrizations in climate models, reducing uncertainties in direct radiative forcing estimates, and enhancing remote-sensing retrieval algorithms. Future work should focus on extending these measurements to additional source regions, and further exploring the effects of dust ageing on optical
properties in dust transport regions.

By providing detailed, source-region observations of dust absorption and scattering, this study strengthens the foundation for more accurate representations of mineral dust in Earth system models, ultimately improving climate predictions and atmospheric radiative budget assessments.

*Code and data availability.* Data and code will be made accessible through an open repository once the publication has been accepted

*Author contributions.* AA, XQ, MK, KK and CPGP designed the measurement campaign. JYD, AA, CGF, AGR, MK, KK, AP, XQ, CR and CPGP participated in the field measurement campaign. CGF processed the meteorological data, CGF and JE treated the PNSD datasets, AP and AGR obtained the airborne and soil mineralogical measurements. JYD processed the optical data-sets, analyzed the results, and summarized and expressed them in this manuscript. JE provided the complex refractive index modelization analysis, analyzed the results,





and summarized and expressed them in this manuscript. All authors provided advice regarding the structure and content as well as contributed

to the writing of the final manuscript.

*Competing interests.* At the time of the research, MR and MI were also employed by the manufacturer of the Aethalometer AE33. At least one of the (co-)authors is a member of the editorial board of Atmospheric Chemistry and Physics.

*Acknowledgements.* The field campaign and its associated research, including this work, was primarily funded by the European Research Council under the Horizon 2020 research and innovation programme through the ERC Consolidator Grant FRAGMENT (grant agreement

no. 773051) and the AXA Research Fund through the AXA Chair on Sand and Dust Storms at BSC. BSC co-authors also acknowledge the support of the Spanish Ministerio de Economía y Competitividad through the HEAVY project (grant no. PID2022-140365OB-I00 funded by MCIN/AEI /10.13039/501100011033 and by ERDF/EU) and the Horizon Europe programme under Grant Agreement No 101137680 via project CERTAINTY. JYD was supported by the European Union's Horizon Europe research and innovation programme under the Marie Skłodowska-Curie Postdoctoral Fellowship Programme, SMASH co-funded under the grant agreement No. 101081355. The SMASH

project is co-funded by the Republic of Slovenia and the European Union from the European Regional Development Fund. JE acknowledges his AI4S fellowship within the "Generación D" initiative by Red.es, Ministerio para la Transformación Digital y de la Función Pública, for talent attraction (C005/24-ED CV1), funded by NextGenerationEU through PRTR. CGF was supported by a PhD fellowship from the Agència de Gestió d'Ajuts Universitaris i de Recerca (AGAUR) grant no. 2020-FI-B 00678. MK received funding through the Helmholtz Association's Initiative and Networking Fund (grant agreement no. VH-NG-1533). KK was funded by the Deutsche Forschungsgemeinschaft

(DFG, German Research Foundation) –264907654, 416816480. We also acknowledge the EMIT project, supported by the NASA Earth Venture Instrument program, under the Earth Science Division of the Science Mission Directorate. We thank Paul Ginoux for providing high-resolution global dust source maps that were very helpful for identifying the experimental site. We also acknowledge Jasper Kok for providing useful insights on our methods.



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
