# Peer review of "Optical Properties and Shape-Dependent Complex Refractive Index Retrievals of Freshly Emitted Saharan Dust"

_EGUsphere, 2025_

## Referee Comment (RC3)

The study "Optical Properties and Shape-Dependent Complex Refractive Index Retrievals of Freshly Emitted Saharan Dust" presents and discusses key optical properties of freshly emitted Saharan dust, utilizing synergies of in-situ measurements conducted close to dust sources in southeastern Morocco.

More specifically, the authors focus on the wavelength- and size-dependent scattering and absorption behavior of dust while they also present a methodology for the retrieval of the imaginary part (k) of the complex refractive index (CRI) while accounting for different particle shape models. It is found that the derived values of k are higher when triaxial ellipsoids are employed in the retrieval process instead of spheres, underscoring the significant impact of particles' asphericity.

I believe the study falls well within the scope of ACP, a novel dataset from a key dust source region is presented which can provide valuable field-based constraints on the radiative effects of dust. The manuscript is well-written / structured, the presentation clear, and the authors give credit to related work. However, I believe there are some key points which require greater clarity and methodological refinement.

**Line 125:** I was a little bit confused with the correction scheme you applied for the truncation error of the nephelometer data. In Section 2.2, you cite Müller et al. (2011), but in Section 2.6.2, you refer to Teri et al. (2022). Could you clarify which method was ultimately used? Additionally, is the PSD derived from the Fidas -based on spherical particles and mean CRI from Di Biagio et al. (2019)- also used here, as described in Section 2.6?

If so, and the Müller et al. (2011) correction (which assumes spheres) is applied alongside a spherical particles assumption for the PSD calculations, how might this influence the accuracy of retrieved scattering coefficients, the asymmetry parameter (g), and the backscatter fraction (BF)? For instance, Gasteiger and Wiegner (2018) suggest that using spherical rather than spheroidal particles can introduce up to 7% uncertainty in hemispheric backscattering.

Lastly, were different correction factors applied during the PM2.5 and PM10 periods? If so, it would be helpful to briefly mention this.

**Lines 249 – 256:** For the calculation of the PSD and k retrievals, i) biaxial prolate ellipsoids with a single aspect ratio, ii) triaxial ellipsoids with the same aspect ratio, iii) a distribution on biaxial prolate ellipsoids with aspect ratios derived from single particle measurements and iv) single irregular dust shapes from Gasteiger et al. (2011) were used.

   i)     How did you derive this specific single aspect ratio value?

   ii)    Should this value be the same in case of triaxial ellipsoids? For example, in Figures 3-5 of Huang et al., (2023) it is shown that the angular distribution of spheroids and triaxial ellipsoids is different for dust particles at side scattering angles.

   iii)   Can you include a plot with the distribution of aspect ratios derived from the single particle analysis during the campaign? How does this compare to the fixed value you have selected in (i)?

   iv)    Why are single particle shapes used here instead of a distribution of shapes? Is it realistic to assume the same irregular shape for all the particles? This question also holds for (i) and (ii).

**Line 419- 420:** *'The analysis for the k retrieval focuses exclusively on periods classified as dust events (see Sect. 3.1) during the PM2.5 measurement period.'*

Why is the PM10 period not included here?

**Section 2.6:** It is confusing to me how shape and k retrievals are disentangled when you need to assume multiple combinations of CRI and shapes to perform the conversion to dust geometric diameters and calculate the PSDs. I assume the combinations of CRI/shapes will be limitless and for a given CRI, a given shape assumption would provide the same results as for another CRI and another shape assumption

**Figure 7:** The maximum differences in the retrieved imaginary refractive index (k) across particle shapes appear to be around 0.001 at shorter wavelengths. How these differences translate into variations in the derived SSA?

Additionally, are the shape-related differences in SSA significant in the context of climate studies? (see for example the study of Mishchenko et al. (2004) on the accuracy requirements for satellite-based SSA retrievals (Table 5)).

**Section 3.3.1:** Regarding the comparison with AERONET, wouldn't it be more appropriate to use the same spheroid shape distribution employed in the AERONET retrieval algorithm to ensure consistency?

Additionally, can you provide the particle size distributions (PSDs) retrieved from the Fidas instrument for the PM2.5 period and compare them with AERONET-derived PSDs? How do the two compare? Could some of the observed discrepancies in the results stem from differences in particle size rather than shape alone?

**Line 119:** *'using the C and multiple scattering fitted values provided therein in a mountain-top station during Saharan dust outbreaks'*

To what extent is it valid to apply a C value derived from a different site with distinct meteorological conditions, instrument state, and aerosol properties (even though in both cases you discuss about dust dominated scenes)? Can you include an uncertainty estimate for the chosen C value (4.82) and discuss how this uncertainty may influence the derived SSA and the retrieval of k?

**Line 153:** *'the aerodynamic diameter of the inlet cut-off, which will vary depending on the inlet cut-off period–PM2.5 or PM10'*

What is the sampling efficiency of the Fidas 200S instrument in relation to particle sizes and wind speed? How these conversions affect the end results for the derived particle size distribution (PSD)?

**Equation 4 and 5:** what is $\beta$ here?

**Line 439:** *'likely more strongly affecting the longer wavelengths'* why is this?

**Line 476:** as a result

**Line 40:** on a microscopic scale

**References** (only those not included in the paper)

Huang, Y., Kok, J. F., Saito, M., and Muñoz, O.: Single-scattering properties of ellipsoidal dust aerosols constrained by measured dust shape distributions, Atmos. Chem. Phys., 23, 2557–2577, https://doi.org/10.5194/acp-23-2557-2023, 2023.

Mishchenko, M. I., Cairns, B., Hansen, J. E., Travis, L. D., Burg, R., Kaufman, Y. J., Vanderlei Martins, and Shettle. E. P.: Monitoring of Aerosol Forcing of Climate from Space: Analysis of Measurement Requirements." Journal of Quantitative Spectroscopy and Radiative Transfer 88 (1): 149–61. https://doi.org/https://doi.org/10.1016/j.jqsrt.2004.03.030, 2004